# Primary radiation damage in bone evolves via collagen destruction by photoelectrons and secondary emission self-absorption

Katrein Sauer [1]✉, Ivo Zizak [2], Jean-Baptiste Forien [3], Alexander Rack [4], Ernesto Scoppola [5,6] & Paul Zaslansky [1,6]✉

X-rays are invaluable for imaging and sterilization of bones, yet the resulting ionization and primary radiation damage mechanisms are poorly understood. Here we monitor in-situ collagen backbone degradation in dry bones using second-harmonic-generation and X-ray diffraction. Collagen breaks down by cascades of photon-electron excitations, enhanced by the presence of mineral nanoparticles. We observe protein disintegration with increasing exposure, detected as residual strain relaxation in pre-stressed apatite nanocrystals. Damage rapidly grows from the onset of irradiation, suggesting that there is no minimal 'safe' dose that bone collagen can sustain. Ionization of calcium and phosphorous in the nanocrystals yields fluorescence and high energy electrons giving rise to structural damage that spreads beyond regions directly illuminated by the incident radiation. Our findings highlight photoelectrons as major agents of damage to bone collagen with implications to all situations where bones are irradiated by hard X-rays and in particular for small-beam mineralized collagen fiber investigations.

X-rays are widely used for imaging as well as for radiation therapy and to destroy unwanted pathogens. The high-energy photons induce ionization that kills cells by breaking down biological building blocks including DNA and protein. X-rays have high penetration power which is why radiation is so useful for sterilization, e.g. for transplants such as corneal xenografts[1] or bone allografts[2,3]. But whenever absorption takes place, scattering and energy deposition will occur. Beyond a certain dose, that depends on the mass and the amount of absorbed photons, radiation damage develops. Crucially, especially for materials such as bone, X-ray operators usually restrict the dose (e.g., ~11 kGy for sterilization[4]), to avoid unwanted side effects of structural damage leading, e.g. to reduced strength[5,6]. Already more than half a century back, reports emerged about collagen protein disintegration and significant structural weakening in X-ray studies of ox-hide[7]. In fact, radiation damage of many collagen-based bony tissues is often identified through observations of reduced mechanical competence. Such degradation is the result of both increased collagen cross-linking as well as fragmentation, for example in human cortical and mouse bones[8–10]. It is known that degradation increases with greater exposure to X-rays[11–13] though it is assumed to decrease with use of smaller X-ray beams[14]. But little is known about the mechanisms leading to radiation damage in bone and in particular how such damage evolves or may affect structural integrity.

Bony materials are ubiquitous in vertebrate skeletons, spanning fish to humans, and samples of such materials are frequently irradiated within the energy range of ~8 to ~80 keV[15–17]. Decades of research

[1]Charité – Universitätsmedizin Berlin, Department for Operative, Preventive and Pediatric Dentistry, Aßmannshauser Straße 4-6, 14197 Berlin, Germany. [2]Helmholtz-Zentrum Berlin, Department for Structure and Dynamics of Energy Materials (SE-ASD), Albert-Einstein-Straße 15, 12489 Berlin, Germany. [3]Lawrence Livermore National Laboratory, Materials Science Division, 7000 East Ave, Livermore, CA 94550, USA. [4]ESRF - The European Synchrotron, Structure of Materials Group - ID19, CS 40220, F-38043, Grenoble Cedex 9, France. [5]Max Planck Institute of Colloids and Interfaces, Department of Biomaterials, Am Mühlenberg 1, 14476 Potsdam, Brandenburg, Germany. [6]These authors contributed equally: Ernesto Scoppola, Paul Zaslansky. ✉e-mail: katrein.sauer@physik.hu-berlin.de; paul.zaslansky@charite.de

employing X-ray diffraction or microtomography revealed important characteristics of bones, with ever increasing temporal and spatial resolutions[8,18]. These have augmented observations by electron microscopy, highlighting textures and lamellar arrangements of the mineralized collagen fibers that define the morphology of different skeletal tissues[19,20]. X-rays easily penetrate these dense biocomposites known to be dominated by carbonated apatite nanocrystals (cAP, with an approximate composition of $Ca_{10}(PO_4)_6COH$[21]). Mineral accounts for 30–40 vol% of bone material, whereas roughly ~20% is water and the rest comprises protein, mainly collagen fibers. All these components are reasonably transparent to commonly used radiation but to different extents, therefore composition differences lead to very different damage consequences. For example, in non-mineralized collagen in dried tendon, damage induced by X-rays is identified following ionization due to exposure to 12 kGy[22]. This is known as "primary radiation damage" and has been shown to split polypeptide chains[23,24] or destroy organic crystals[14]. On the other hand, hydrated tendon samples exhibit degradation only after exposure to ~20 kGy. This suggests that water slows-down collagen ionization or shields the fibers from direct damage, despite free radical formation by radiolysis[25,26]. When this occurs, reactive species induce structural degradation known as "secondary radiation damage", mediated for example by mobile OH-radicals[2,24,27]. Yet the higher doses sustained by wet collagen, at least initially, suggest that overall, the tissue is less damaged and that a smaller portion of the impinging radiation energy actually directly ionizes and fragments the fibers in the material.

Radiation damage to collagen with and without hydration is fairly well understood, however, the situation becomes more complex within nanocomposites of mineralized collagen fibers. Studies have found that the solubility of collagen in irradiated dry bone is 10× higher than the solubility of wet bone samples[28], indicative of a protective or cross-linking role that water may have in the latter samples. The heavier mineral elements, calcium (Ca) and phosphorous (P), absorb most of the incident high-energy radiation and should therefore better shield collagen than water does. But this does not seem to be the case: hydrated mineralized mouse bone samples exhibit fragmentation at doses as low as ~5 kGy[10], far lower than observations of doses sustained in non-mineralized tendons. Clearly, the cAP nanocrystals of bone do not protect collagen from damage, as might be expected from simple assumptions of absorption.

There have been reports of radiation damage from X-ray fluorescence due to the presence of heavy elements embedded in organic matrices within polymers, leading to P–O and C–O bond breakage[29]. Such interactions have been described even for cryo-cooled macromolecular crystallography radiation studies[30,31]. Gradual fading of X-ray diffraction (XRD) reflections has in fact served as an indicator for organic crystal degradation correlated with increasing radiation damage of molecules located in the X-ray beam path[14,32]. Precise determination of absorbed doses[33] as well as photoelectron escape and fluorescence[34] have made it possible to predict radiation damage in such experiments[14,35]. However, these kinds of predictions are not yet available for bony material, where cross-linking and cleavage are both observed in collagen fibers.

To better understand X-ray interactions with skeletal tissues, the effects of photoelectrons and X-ray fluorescence must be considered, phenomena that have largely been overlooked in radiation studies of bone. Secondary radiation excitation by heavy elements such as Ca and P can lead to cascades of emission and absorption[36], since scattered electrons and fluorescence excite lower energy absorbing adjacent elements[37].

In this work we make use of the advantages of microfocus X-ray beams to initiate and quantify direct observations of structural degradation and the break up of collagen. To determine damage within fully mineralized bone, we combine in situ observations by second harmonic generation confocal laser scanning microscopy with electron microscopy and simulations. We identify fragmentation of the collagen backbone and we quantify damage accumulation, combining micro structural imaging with indirect residual stress relaxation identified within cAP mineral nanocrystals. Flux, absorption, and exposure time are the major dominating parameters, suggesting that standard dose-to-sample calculations incompletely predict structural damage in bones studied with smaller cross-section X-ray micro beams. Damage in dry bone material is enhanced through cascades of self-absorption, fluorescence, and electron interactions originating in the mineral nanocrystals. This occurs in addition to any secondary radiation damage and collagen cross-linking that is expected in hydrated samples. Our studies show that primary radiation damage in bone spreads non-linearly, increasing with time. It spans multiple micrometers outside μm-sized irradiated sites, with collagen fragmentation accumulating from the onset of exposure to the incoming hard X-rays.

## Results

### Primary radiation damage effects revealed in bone

In situ observations by second harmonic generation (SHG) images of collagen in irradiated bone revealed damaged zones following exposure to X-rays during commonly performed experiments. Collagen oriented across the image plane gives rise to an SHG signal that is proportional to the density of structurally intact collagen fibers. Figure 1a schematically illustrates two typical experimental scenarios. In the first example, entire sample regions are imaged by micro-computed tomography (μCT). In the second example, points are irradiated by a microfocus beam for XRD measurements. Comparison between SHG intensity maps of fish bones observed by SHG in samples imaged both before and after irradiation reveals distinct dimming of the collagen fiber textures. Such dimming is seen following experiments of μCT (Fig. 1b), in measurements by XRD-μCT (Fig. 1c), and even in regions exposed to single XRD (Fig. 1d) shots.

Example doses absorbed during three typical experiments are given in Table 1 (for dose calculations, see Supplementary Notes and Supplementary Table 1). The observed damage, however, is not uniform, as indicated by yellow arrows highlighting affected zones identified by SHG. Due to the requirement of repeated irradiation, μCT scanned regions absorb doses that are an order of magnitude higher than doses typical for common XRD experiments. With increasing exposure, all experiments lead to notable damage (dimming, fading) of the collagen SHG signature. Concomitantly, the samples tend to become more stiff and brittle to handle.

### Quantification of radiation damage spread in bone

The visibility of collagen in μm-sized sites analyzed by SHG makes it possible to directly track primary radiation damage development. This is best observed when samples are placed 2 cm behind a 20 μm-diameter pinhole (for pinhole and beam dimensions see in Supplementary Table 2) that defines the geometry of our 18 keV X-ray beam (Supplementary Fig. 1). Dry bone specimens of pike fish reveal reduced collagen integrity and dimmed intensity following increased exposure to the incoming X-ray beam. Figure 2 shows corresponding SHG and backscatter imaging scanning electron microscopy (SEM-BEI) of a piece of cleithrum bone irradiated for four different exposure times (40 s, 80 s, 160 s, and 320 s). Despite the moderate flux of the incident beam ($5.5 \times 10^7$ ph s$^{-1}$), clear signs of radiation damage are visible in regions irradiated for as little as 40 s (black spots, Fig. 2a), exhibiting increasing prominence with longer exposure times. The loss of SHG intensity is indicative of collagen destruction and likely burn-off. The exact same spots appear with inverse contrast (bright) when imaged by SEM-BEI (Fig. 2b). Collagen burn-off indeed leads to an apparent higher relative mineral density, observed as increased brightness in the SEM-BEI images. With longer X-ray exposure times (40 s to 320 s) the damage becomes more visible by both SHG and SEM-BEI (lateral extents determined for this damage are given in Supplementary Table 3).

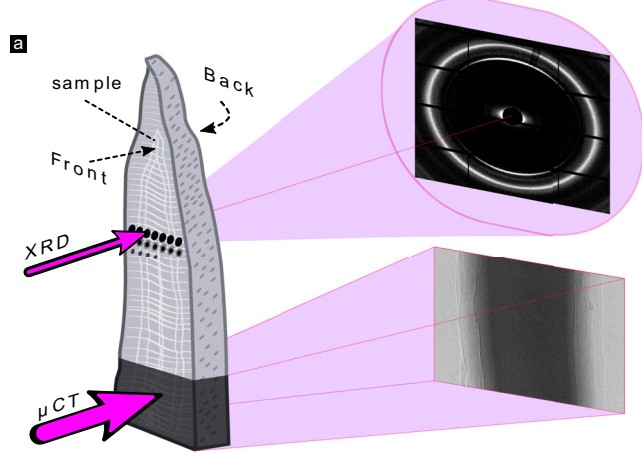

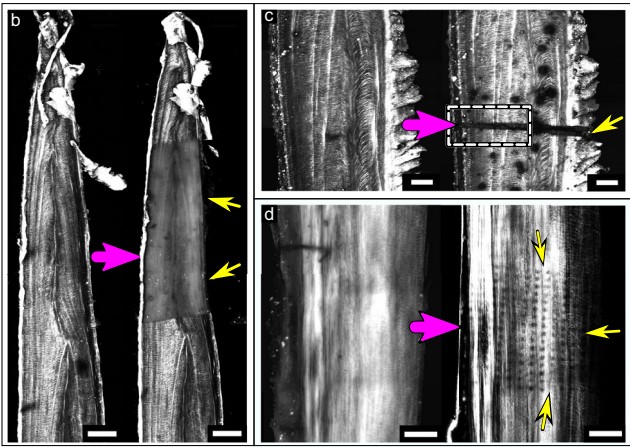

**Fig. 1 | Sources of radiation damage to collagen in fish bone exposed to different synchrotron experiments. a** Schematic illustration of X-ray diffraction (XRD) and micro-computed tomography (μCT) experiments. Incoming radiation indicated in magenta (**b**) SHG images reveal collagen distribution and burn-off in samples of pike fish bone imaged before (left) and after (right) a μCT experiment performed at ID19, ESRF (*n* = 39 samples). Scale bar: 250 μm. **c** Before (left) and after (right) an X-ray diffraction μCT (XRD-μCT) experiment performed at the mySpot beamline, BESSY (*n* = 1 sample). Scale bar: 150 μm. **d** Before (left) and after (right) a 2D mapping XRD mySpot experiment (*n* = 25 samples). Scale bar: 200 μm. Areas of damage appear dark (identified with yellow arrows) with reduced SHG signal due to damage to the collagen fibers.

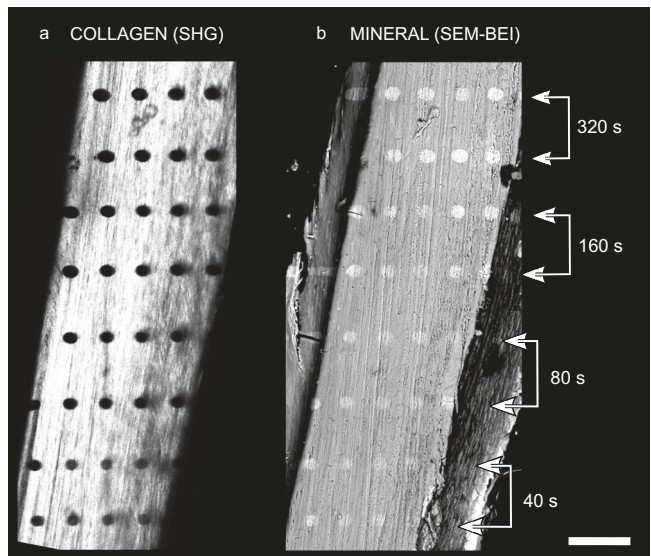

**Fig. 2 | XRD induced radiation damage manifestation: second harmonic generation (SHG) versus backscatter imaging scanning electron microscopy (SEM-BEI). a** SHG image of irradiated pike bone. The SHG signal vanishes in sites where the beam impinges on the sample (black oval areas (*n* = 7 samples)). **b** SEM-BEI image of irradiated pike bone. The same imprints of damage in **a** appear brighter in the electron microscope (*n* = 2 samples). The scale bar is 100 μm.

The SHG is useful to demonstrate the presence of damage both on the entry side of the incident X-ray beam ('Front', see schematic in Fig. 1a) and on the exit side ('Back') of the bone samples (Supplementary Fig. 2 and Supplementary Table 4). The visibility of damage found on the 'Back' is however somewhat diminished compared to the 'Front', mainly observed in regions exposed for shorter times (80 s). Indeed, collagen on the 'Back' is more protected from burn-off because the beam intensity diminishes due to absorption while propagating through the ~300 μm bone thickness. In fact, direct measurements of X-ray transmission through the sample find a ~25% decrease in beam intensity. As compared with higher-density mammalian bones, fish bones have only a modest mass density of ~1.4 g cm⁻³ (see Supplementary Notes 3.1) corresponding to a mineral density of ≈1 g cm⁻³[38] which significantly attenuates the beam propagating through the bony material.

We demonstrated burn-off damage to collagen in a range of bony tissues. Figure 3 compares radiation damage in four different species (pike cleithrum, bovine tooth, mouse tibia, and pig jaw) following different X-ray exposure times (40 s, 80 s, 160 s, and 320 s). For each sample, the damaged area increases with increasing exposure (Fig. 3, from top to bottom). Damage extents based on SHG measurements for the different bones irradiated at 40 s and 320 s are listed in Supplementary Table 5. Additional examples for damage in pike bone exposed to different beam sizes and beamline configurations (Supplementary Fig. 1e and f) are presented in Supplementary Figs. 3, 4, and 5. In all cases, the pike fish bone damage imprint is notably oval, with damage extending sideways, orthogonal to the highly aligned bundles of collagen fibers. In the other, more complex and denser bony tissues, the damage is more uniformly distributed around each radiation site, with only hints to damage extension into irregular bony textures.

For all the beam sizes investigated, the damage accumulates and spreads beyond the X-ray irradiated area, expanding the damage imprint over time. With known sample thickness, and from direct measurements of the lateral extent of damage, it is possible to determine the damage-affected bone volumes. In fact, in our experiments, the ratios of illuminated to damaged volumes (relative damaged volume) are identical to the corresponding ratios of the X-ray damaged and X-ray irradiated cross-sectional areas (due to the ~300 μm uniform sample thickness) and these ratios are shown in Fig. 4. The damaged cross-sectional area expands as a function of exposure time, as seen for a range of different beam sizes (Supplementary Fig. 4). The non-linear trend and saturation observed when the data are normalized by the beam size and plotted as a function of dose, can be modeled by first-order exponential fits (Supplementary Notes 3.3). Though the rate of damage accumulation decreases with decreasing beam sizes, the relative damaged volume increases. Thus it can be seen that irradiation of bone samples with a 5 μm versus a 100 μm beam for 320 s produces a relative damaged volume of ~260% versus ~120%, respectively. Therefore in bony materials, different from organic crystal

## Table 1 | Calculated dose of the experiments shown in Fig. 1

| Experiment | μCT | XRD-μCT | XRD |
|---|---|---|---|
| Dose [kGy] | 2689 | 3052 | 118 |
| Synchrotron | ESRF | BESSY | BESSY |

Calculations are based on ref. 8, see Supplementary Notes 3.1 and 3.2 and are the doses used to induce the damage observed by SHG in μCT in Fig. 1b, XRD-μCT in Fig. 1c, and XRD in Fig. 1d.

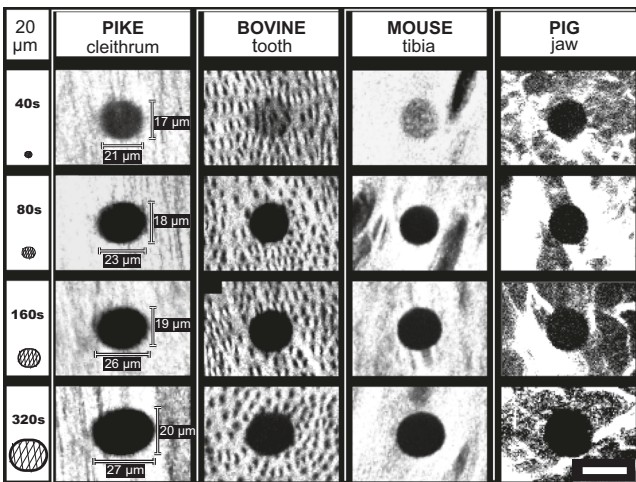

**Fig. 3 | Comparative SHG imaging of radiation damage across different bony tissues with increasing exposure.** The SHG shows clear signs of collagen disruption in the 'Front' of pike cleithrum, bovine tooth, mouse tibia, and pig jaw with increasing radiation exposure from 40 s to 320 s (dark circular areas). Irradiating at 40 s reveals the smallest area of damage with notable dimming of the SHG intensity, particularly well visible in the bovine and mouse samples. With increasing exposure time, the lateral extent of radiation damage increases and becomes notably oval orthogonal to the collagen texture direction. This experiment was repeated 7 times on pike bones, 2 times on bovine teeth, 2 times on mice tibia, and 3 times on pig jaws. Scale bar: 20 μm.

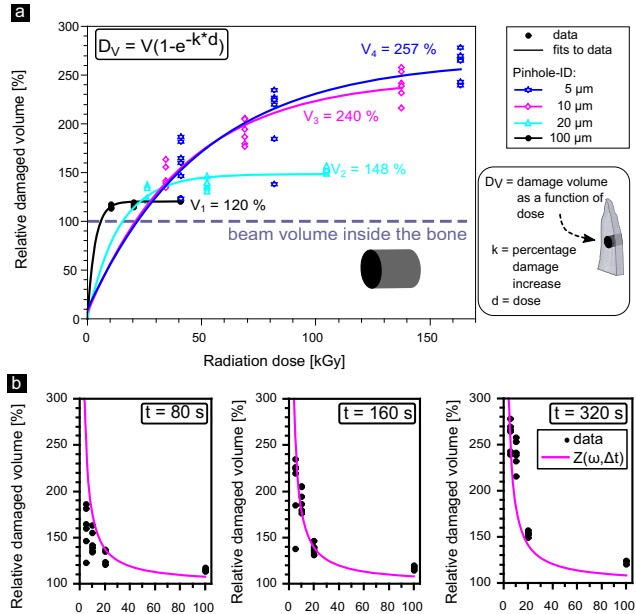

**Fig. 4 | X-ray damage in collagen as compared to the lateral beam size.**
**a** Accumulated damage ($D_V$) as a function of radiation dose: with decreasing lateral beam diameter (Pinhole-ID: 100 μm (black rectangles), 20 μm (cyan triangles), 10 μm (magenta diamonds), and 5 μm (blue stars)) the damage extent (V) increases relative to the beam size (=100%) to ~120% ($k = 0.33\%$ s$^{-1}$), ~150% ($k = 0.07\%$ s$^{-1}$), ~240% ($k = 0.02\%$ s$^{-1}$) and ~ 260% ($k = 0.02\%$ s$^{-1}$), respectively, after exposure for 320 s. In all cases, the damaged volume substantially exceeds the beam volume inside the bone (indicated with a line parallel to the x-axis). **b** Accumulated damage when irradiated for different exposure times (*t*): 80 s, 160 s, and 320 s with a 20 μm pinhole. Black dots refer to relative damaged volume with respect to the beam size, as determined by SHG. Magenta lines depict the results of the model function $Z(w, \Delta t)$ for the case of $\zeta_V \approx 1$.

experiments, the smaller the beam size, the more severe the relative damage volume becomes. This has important implications when adjacent points are to be measured.

Our direct observations of collagen destruction suggest that estimations of radiation damage based only on incident radiation dose and time underestimate the extent of damage induced in the bony samples. The composite bone nano-architecture must be considered to better understand the different interactions of X-ray radiation with the fibers versus the cAP crystals. Upon exposure to radiation, a cascade of electron and subsequent X-ray fluorescence (XRF) photons are generated, mainly in the higher-absorbing elements in the cAP nanocrystals (Supplementary Notes 3.4 and Supplementary Figure 6). The magnitude of this secondary emission is proportional to the primary beam intensity, its energy, the photoionization cross section, the fluorescence yield and the radiative rate[37,39]. Once excited, electrons and XRF photons propagate across the sample in all directions until they are absorbed. This leads to radiation beam energy being deposited in sample regions beyond those directly illuminated by the incident beam. The main absorber is Ca in cAP, which, upon excitation, emits Ca K-shell electrons (Supplementary Figure 6). Though P in cAP certainly also contributes, we concentrate on the former, since the ejected Ca K-shell electrons are the most abundant. They have energies that are higher than emitted XRF photons (e.g., with a 18 keV primary beam, for each 4.0396 keV absorbed photon emitting XRF—see Supplementary Fig. 6—an additional 13.964 keV electron is ejected). In this manner, the illuminated cAP crystals become a source of high-energy electrons. Monte Carlo simulations revealed the propagation depths of such photoelectrons in different bone specimens with compositions consistent with experimental observations (see Supplementary Fig. 7 and Supplementary Notes 3.4). A probability distribution function ($P(d_e)$) of the penetration depth of electrons ($d_e$) along any line radiating outward from the beam center into the surrounding bone matrix can be extrapolated (Supplementary Fig. 8). Due to the geometry of the electron source (i.e. with respect to the center of the incoming circular X-ray beam), the function $P(d_e)$ is symmetric such that:

$$P\left(+\left|d_e\right|\right) = P\left(-\left|d_e\right|\right). \tag{1}$$

In order to understand how far the energy carried by photoelectrons is deposited into the surrounding bone matrix, we simplify the analysis to focus on a 1D function that represents the photoelectron source along a horizontal line *B* created within the area illuminated by the incident X-ray beam such that:

$$B(w, x) = \text{rect}\left(\frac{x}{w}\right) \tag{2}$$

with *w* corresponding to the X-ray beam width and *x* representing the distance from the center of the source. By convolving $B(w, x)$ with $P(d_e)$, we obtain $\eta(w, x)$ representing the energy deposition region with respect to the source:

$$\eta(w, x) = \frac{\int_{-\infty}^{+\infty} B(w, \xi) P(x - \xi) d\xi}{\int_{-\infty}^{+\infty} P(\xi) d\xi} \tag{3}$$

where normalization by $\int_{-\infty}^{+\infty} P(\xi) d\xi$ ensures energy conservation. The damaged region is time-dependent, since increasing numbers of ejected photoelectrons will increase the amount of energy deposited. Therefore by integrating $\eta$ over time we obtain a time-dependent integral *H*:

$$H(w, x, \Delta t) = \int_{0}^{\Delta t} \eta(w, x) dt \tag{4}$$

with $\Delta t$ representing the exposure time. The larger the exposure time, the more we observe effects of the ejected photoelectrons in the surrounding bone matrix.

We now define $x_w = x_w(\Delta t)$ such that

$$H(w, x_w, \Delta t) = 0.05 \qquad (5)$$

with 0.05 representing an energy threshold beyond which damage is observed in the bone. We can now calculate the relative damage expressed as the ratio between the damage diameter and beam diameter along a line extending outwards from the center of the beam $\zeta(w, \Delta t)$, defined as:

$$\zeta(w, \Delta t) = 2\frac{|x_w(\Delta t)|}{w}. \qquad (6)$$

The sample thickness in which the beam propagates is equal to the sample thickness in which radiation damage develops. Therefore, the ratio of damage volume and beam volume in the sample is identical to the ratio between the elliptical damaged area and the beam cross-sectional area. This ratio we define as the relative damage volume, which is identical to the product of the $\zeta$ functions for the orthogonal horizontal and vertical directions ($\zeta_H$ and $\zeta_V$), such that:

$$Z(w, \Delta t) = \zeta_H(w, \Delta t) \cdot \zeta_V(w, \Delta t). \qquad (7)$$

For the case of the highly anisotropic pike bone samples shown in Fig. 4, $\zeta_V \approx 1$ therefore $Z(w, \Delta t) \approx \zeta_H(w, \Delta t)$), since the relative damage along the vertical direction is negligible as compared to the horizontal relative damage (see Supplementary Tables 4, 5). The relative damaged volume can be simply calculated by means of Equation (6).

Examples of $B(w, x)$, $H(w, x, \Delta t)$, and $\eta(w, x)$ are provided in Supplementary Figure 9, demonstrating how the relative electron spread is greater for the smaller 5 μm beam diameter. Values of $Z(w, \Delta t)$ for a range of beam diameters (100 μm and smaller) and three exposure times (80 s, 160 s, and 320 s) are given in Fig. 4b (magenta lines) and compare well with experimental values obtained by analysis of the SHG images of damage in pike bone (black dots). Supplementary Table 6 lists some predicted relative damage values for several exposure times for increasingly smaller beam sizes. Our data predict that when decreasing the beam size, the relative damage will increase non-linearly, exceeding 150% for beams smaller than 20 μm. Thus, for a beam sized 1 μm, we expect ~×10 more relative damage and for a beam of ~0.1 μm we expect damage to be ×75 larger than the beam diameter. One consequence of these predictions is that measurements in bone of adjacent points should be sufficiently spaced, to minimize biasing effects of accumulation of radiation damage, which clearly depends on how far the electrons scatter through the bony nanocomposite and not on the beam diameter. Note the impressive agreement between the experimental data and the model, although only the horizontal orientation is integrated. This is striking because only primary photoelectrons ejected from Ca are considered here, ignoring photoelectrons emitted from other elements. Additional details are provided in Supplementary Notes 3.4.

Further confirmation to the damaging role of photoelectrons emitted from cAP is obtained by comparing damaged sizes in mineralized versus demineralized samples (Fig. 5). In mineralized pike bones, the damage cross-section is larger than in demineralized samples, despite similar exposure to radiation. Figure 5a shows an example SEM-BEI image of damage imprints (bright spots). Demineralized samples reveal dark damage spots presumably due to collagen burn-off that appears black in the electron microscope (Fig. 5b). SHG shows dark spots in both samples, indicative of collagen destruction in both cases. But the size of damage in mineralized pike bone is ~5 μm larger (horizontally: 27.16 ± 0.72 μm and vertically: 22.13 ± 0.41 μm) as

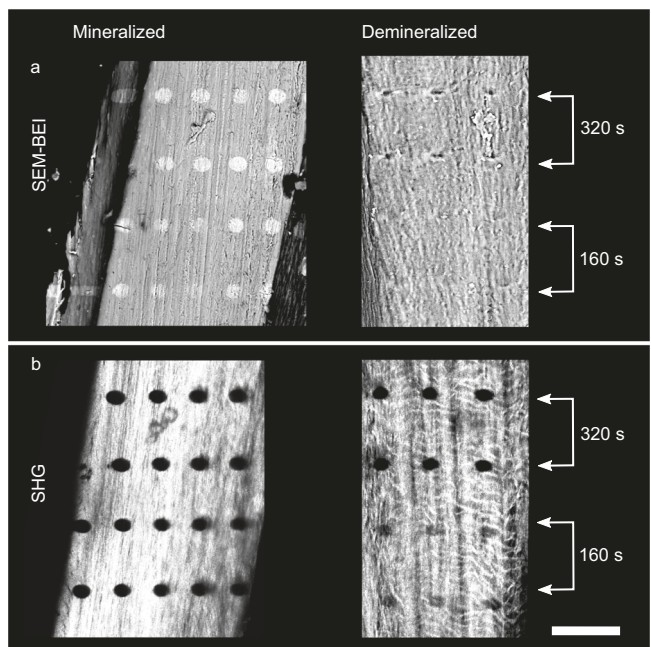

**Fig. 5 | Damage is visibly larger in mineralized versus demineralized bones exposed to identical radiation. a** Backscatter imaging scanning electron microscopy (SEM-BEI) images of irradiated sample with mineral show bright damage imprints. Damage imprints in demineralized samples appear dark in SEM-BEI images and are smaller in demineralized samples compared to mineralized ones ($n = 2$ for mineralized bone and $n = 3$ for demineralized bone). **b** Second harmonic generation (SHG) images of samples shown in (**a**) demonstrate how irradiated bone containing mineral reveals black spots indicative of damage. Damage imprints (black) in demineralized bone appear smaller compared to mineralized bone samples. The scale bar is 100 μm. XRF sum spectra of the native and demineralized samples are shown in Supplementary Figure 6, identifying abundant calcium (Ca) in the former and no Ca signal in the latter. Strontium (Sr) is a natural substitute to Ca and is also removed upon demineralization.

compared with the damage observed in demineralized samples (horizontally: 22.49 ± 1.43 μm and vertically: 16.24 ± 0.61 μm). Note that in both cases we used configuration 1 of the beamline and a 20 μm pinhole, with the exact same exposure times. Consistent with the previous considerations, the absence of cAP in the demineralized samples lowers the probability of producing photoelectrons and secondary emissions, resulting in far smaller lateral extension of the damaged area.

## Mineral nanoparticles as indirect radiation damage sensors

SHG is unable to reveal early onset of radiation damage. We therefore make use of the tight interactions between the mineral particles and dehydration-tensed collagen fibrils known to compress cAP in desiccated bone tissues. The dried collagen protein fibers induce a significant shrinkage of the apatite crystals, leading to residual strains[40–42] (Fig. 6a). XRD reveals the (002) diffraction peak corresponding to the c-lattice parameter of the cAP nanocrystal, that is more or less co-aligned with the collagen nanofibrils. Peak position analysis of the XRD patterns for different bone samples (pike, bovine, mouse, and pig) makes it possible to calculate the relative strain ($\Delta c/c$) as a function of the exposure time, as shown in Fig. 6b (see Supplementary Table 7 for typical XRD determined bone mineral characteristics).

In all the dried bone samples that we measured by XRD, the initial c-lattice parameter increases with increasing X-ray exposure time, revealing stress relaxation. Strain is released gradually as the collagen loses the capacity to sustain the dehydration-induced compression stress in the mineral nanoparticles (Fig. 6a). Comparison of residual strain release rates (percentage-change) for three different photon

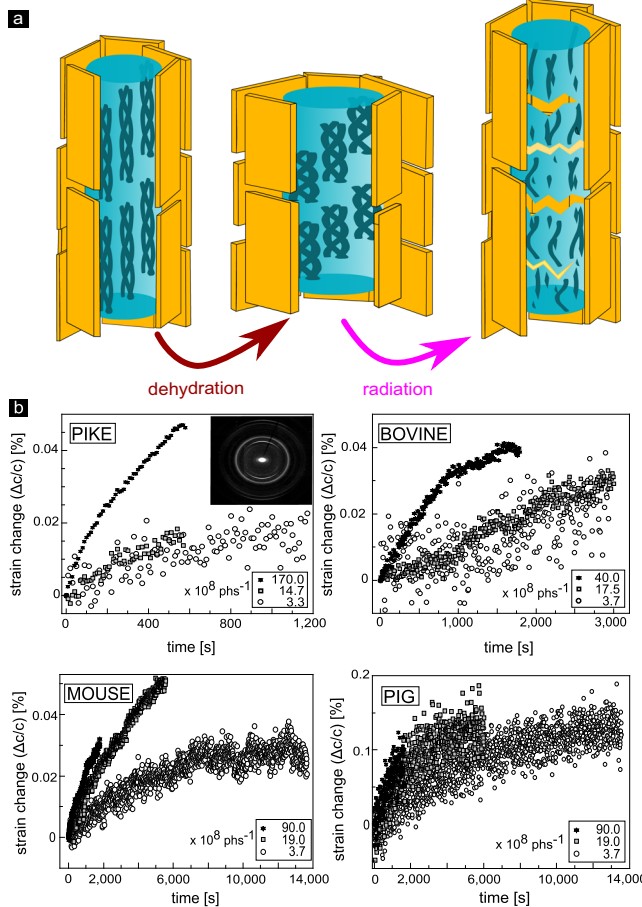

**Fig. 6 | cAP nanocrystal deformation and residual strain as indicators of radiation damage accumulation in the bone bulk. a** Schematic illustration of a model of mineralized collagen fibers and the effects of dehydration (shrinkage, marked with brown arrow) and radiation damage (expansion, marked with magenta arrow) of the nanocrystals. Collagen shown here (triple helices within cyan pillars) runs vertically and is co-aligned with the tightly-attached axially aligned mineral cAP crystals (platelets, yellow). Together, both ingredients establish the basic building blocks of bony tissues: mineralized collagen fibers. When dehydrated, collagen condenses and due to the tight attachment, the co-aligned apatite mineral cAP crystals become compressed along the c-axis[40]. Following X-ray irradiation and ionization, collagen molecules become fragmented and lose the capacity to compress the cAP crystals. **b** Percentage strain change within pike cleithrum bones, bovine teeth, mouse tibia, and pig jaws measured with different photon fluxes (photons per second (phs⁻¹)). For all XRD measurements, the energy was 18 keV. With accumulation of damage, the collagen backbone can no longer sustain residual compressive stress in the mineral crystals, and consequently the c-lattice parameter increases. The diffraction pattern shown for pike highlights the highly anisotropic arrangement of mineralized fibers with axial alignment of the (002) Debye rings producing upper and lower arcs.

fluxes, indicates that relaxation commences seemingly from the moment of radiation onset and appears to decrease and slow down with time. Note that the lower flux yields extremely weak XRD patterns and thus requires longer exposure times and results in fewer data points. Longer exposures are thus needed to reach sufficiently strong diffraction patterns in which the atomic d-spacing of the c-lattice can be reliably determined from the (002) reflections, yet the trends in the data are similar. Note also that for all fluxes, the initial damage accumulation rate is higher than at later times. The early rapid increase in relaxation is always followed by slower strain relaxation rates, indicative of a slowing down of the accumulation of radiation damage in collagen.

Our XRD stress relaxation data match observations by SHG, where collagen fibrils appear to accumulate damage (Fig. 3) and disintegrate. With higher flux, collagen disruption rates increase, seen as a rapid relaxation of the c-lattice parameters in different bony tissues.

We conclude that radiation damage accumulation in the bony nanocomposite is non-linear, and that the affected damaged volume increases with time.

## Discussion

Our data show that primary radiation damage in bone is strongly related to photoelectron scatter cascades and damage to collagen. Photoelectron cascades are generated by excitation of heavier atoms (e.g. Ca) by both incoming and fluorescence X-rays. These secondary radiation processes induce structural disruption in the collagen network, and they spread out beyond the volume directly illuminated by the primary beam, exhibiting increasing burn-off with increasing exposure time.

SHG reveals expansion of radiation damage, clearly visible on both the 'Front' and the 'Back' of the irradiated bones observed within a variety of samples from different animals. The lack of cells or any lacunar-canalicular inhomogeneities[38,43] of pike fish and the highly anisotropic elongated clusters of mineralized collagen fibers in these bones are useful to ascertain that X-ray interactions occur within reasonably uniform bone nanocomposite material comprising almost exclusively mineral nanoparticles and collagen fibers. Other bony tissue, such as bovine-teeth, mouse tibia, and pig-jaw are somewhat denser as compared to fish bone and they also include multiple structural inhomogeneities such as channels and cells. These texture and density differences affect electron scatter trajectories. Yet all measurements show the same trends in time-dependent damage expansion observed beyond the illuminated volume, especially noted with decreasing beam sizes. By dehydration, we confine our analysis to primary radiation damage, avoiding radiolysis. Consequently, our work complements previous studies that showed secondary radiation damage and collagen degradation often linked to fibril cross-linking[22].

Second harmonic generated signals arise from electric dipole moments of collagen fibrils[44–47]. Collagen Type I mainly consists of Glycin ($C_2H_5NO_2$), Proline ($C_5H_9NO_2$), and Hydroxyproline ($C_5H_9NO_3$)[48]. When linked into long-chained molecules, they form tropocollagen, an alpha triple helix made of three of those chains. Stacked tropocollagen stabilized by intermolecular cross-links[5] possess non-centrosymmetric properties on the micrometer length-scale. Collagen birefringance and high degree of order are necessary conditions for SHG imaging. When the ability to produce optical SHG signals is lost, this is due to structural breakdown of collagen[49–51]. As a direct result of ionization, SHG signals are either diminished or completely extinguished appearing dark in the radiation exposed regions, indicative of collagen burn-off. These results are further confirmed by SEM-BEI, where the electron backscatter signal in mineralized samples increases, as the relative mineral density increases, appearing brighter on the bone surface (Figs. 2, 5).

We observe that in bone, radiation damage, breakdown of the collagen backbone (Supplementary Fig. 10) and the relative damaged volumes are not linear functions of dose, which is different to observations in studies of protein lysozyme crystals[14,23].

In bone, we find that radiation damage develops initially very rapidly after which it progressively accumulates in the bone matrix exceeding the dimensions of the incident beam. In good agreement with modeling, our experimental results show how electrons distribute and enlarge the extent of damage. For a range of incoming beam diameters, the damage zone exceeds the beam-irradiated volume with smaller beam sizes producing larger relative damaged zones. Yet fiber and tissue texture and mineral density both affect how the damage

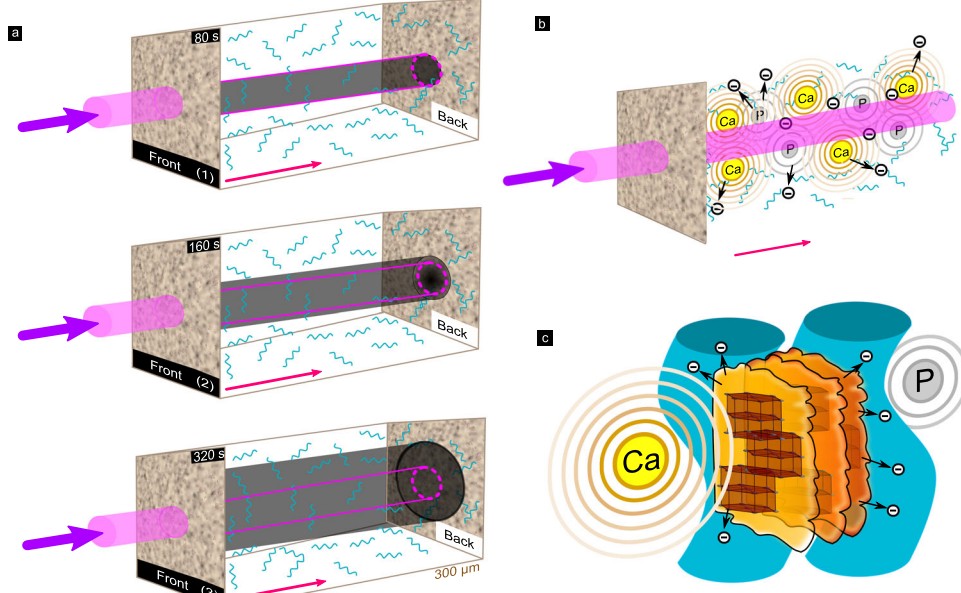

**Fig. 7 | Schematic representation of radiation damage expansion in bone.**
**a** Macroscopically, exposure to an incident X-ray beam (magenta) during typical XRD experiments will result in increased damage as a function of exposure time: (1) 80 s, (2) 160 s, and (3) 320 s. Following irradiation of the bone (beige), collagen becomes increasingly damaged (dark gray) and the damage is most intense in the center of the entry site of impingement on the bone sample. Damage expands laterally with increasing exposure time. At lower irradiation time (80 s), damage is smaller, but with increasing exposure the size grows (160 s), to significantly exceed (320 s) the lateral beam diameters. Though collagen absorbs only a small portion of the incident intensity, calcium (Ca) and phosphorous (P) in the bone absorb and emit significant photoelectrons and fluorescence as the beam propagates through the specimen. Due to absorption, the beam intensity decays towards the 'Back' of the sample. **b** At the ultra-structural level, a cascade of electron scattering (black circles with minus symbol) and fluorescence is created, dominated by Ca (rings) delivering energy that is highly absorbed by both P and the organic components of the bone (cyan). Secondary emission by P and other sources of ionization will lead to accumulation of broken backbone C–C and C–N bonds. The Ca photoelectrons scatter many micrometers beyond the rims of the incident beam. **c** At the level of crystals and collagen fibers, the incoming radiation excites Ca and creates secondary radiation sources in the bone with an attenuation length of several μm. Excitation of the elements comprising the mineral nanoparticles (brown/yellow), results in ejection of electrons (black circles) that scatter in the matrix (attenuation length ~1.5 μm). The combined effects of lower energy fluorescence and secondary electron scattering are main contributors to damage of collagen in the matrix (illustrated as cyan wavy pillars).

spreads. Thus, the widely used approach for predicting damage in bone that is based only on the absorbed dose becomes increasingly inaccurate when using beam sizes down to and beneath the micrometer length scale.

Our results further raise concerns regarding the use of gamma irradiation for sterilization of bone samples, despite minimal indications for degradation of the mechanical properties[52]. While there have been proposals for doses that may be safe for sterilization radiation[53], we posit that there is likely no truly safe dose for bone material exposed to hard X-rays, and that absorption will always result in an inevitable cascade of electron scattering with likely damage to the organic fibers. Such damage may however be resolved by the normal processes of bone remodeling in vivo[46].

In SHG, damage areas in both mineralized and demineralized bones are visible as dark spots. SEM-BEI showed that demineralized irradiated spots appear dark, unlike their bright mineralized counterparts. In demineralized pike bone, both SHG and SEM-BEI show a substantial smaller lateral damage extent compared to mineralized bones. This serves as further evidence that photoelectrons are major causes of collagen degradation in X-ray irradiated mineralized bony tissues.

Photoelectron emission will occur due to the high abundance of Ca (and P) in the mineral crystals. This is due to the electron binding energies in the ~2–4 keV range for these elements, that possess a substantial photoelectron ionization cross-section. Due to localized electron scattering out of the mineral crystals and cascades of emitted and absorbed low-energy fluorescence (See Supplementary Notes 3.4) radiation damage both accumulates and grows with time. Consequently, ionization of the collagen surrounding the mineral is probably

unavoidable. Therefore, for estimation of radiation damage, it is insufficient to consider the low absorption of photons when using high-energy X-rays, because the incident radiation is only partially responsible for the degradation in collagen. Multiple secondary interactions will create scattered electrons that can break down covalent C–C and C–N bonds in the collagen backbone extending beyond regions directly illuminated by the incident beam. Importantly, electrons ejected inside the bone cAP crystals will have sufficient energy to scatter through the crystals in which they are created, as Monte Carlo simulations show (Supplementary Fig. 7).

Our model in Equations (3)–(7) explains why the smaller the beam diameter, the larger the relative damaged volume expected. $P(d_e)$ depends on the electron energy but not on the beam size. $B(w,x)$ is beam size-dependent, such that the smaller $w$, the larger the effect of convolution by $P(d_e)$. Broadening or smearing effects of the primary beam (e.g. divergence due to different sample positioning with respect to the source, see Supplementary Fig. 1f and Supplementary Fig. 5) will affect the relative damaged volume distribution, but will not change the fundamental interactions and damage spread via photoelectron excitation.

Our proposed mechanism of collagen breakdown in bone domains irradiated by X-ray beams is outlined schematically in Fig. 7.

The finite size of the incoming beam typical of XRD measurements (Fig. 7a, magenta) produces damage with an imprint that expands over time. Along the beam path through the bone, energy is absorbed as the incoming beam loses intensity towards the 'Back', creating photoelectrons and secondary cascades. With increasing exposure time, more surrounding collagen becomes disrupted (dark gray areas), as the damaged volume increases (Fig. 4). In the volumes

directly irradiated by the incident beam, secondary sources overlap with cascades of absorption and excitation extending beyond the beam periphery (Fig. 7b). The main drivers of damaging self-absorption and photoelectron excitation are Ca and P (Fig. 7c). Due to the high concentration of these heavy elements, especially in the crystalline domains in the mineral, secondary interactions of fluorescence and electron scattering expand the damage to include collagen fibers surrounding the bone crystals up to several μm away from the impingement site of the incoming X-ray beam (collagen fibers schematically indicated by wavy lines). At the same time, the bone ingredients, mainly mineral, attenuate the beam as it propagates along the irradiation path through the sample.

Strain evolution of the *c*-axis of cAP observed by XRD revealed that damage ensues at early stages of X-ray radiation, though it must accumulate to have a quantifiable effect on strain release. The observed non-linear change in the c-lattice parameter of the nanocrystals as the exposure time increases, highlights the dynamics of damage accumulation in the beam path, as diffraction is only obtained from the X-ray illuminated bone volume. The crystals in such experiments act as strain sensors within the tissue[17,40,54,55]. Dehydration prior to irradiation in our experiments caused collagen to tense-up and shrink and so compress the tightly attached crystals in the bony tissues[40–42,54,55], particularly along the c-axis of cAP. Our experimental findings of increasing c-lattice parameters with increasing flux and time are in good agreement with previous measurements in bovine bone[56], showing reduced residual strain with increased radiation exposure. Those and our own findings in fish bone, bovine teeth, mouse tibia, and pig-jaw suggest that early rapid damage to collagen nucleates and grows at a rate that gradually slows down. The rate of relaxation of strains in the pre-stressed mineral crystals decelerates as the remaining intact collagen continuously degrades, so that the organic fibers lose the capacity to sustain the dehydration-induced compressive residual stresses. One outcome of this is that any measurements of bones by high-energy X-rays are likely to lead to collagen damage which would impact lattice measurements in bone samples in which residual stresses exist. This effect needs to be considered when XRD is used to measure bony tissue cAP lattice spacings, at least in dry bone specimens.

A surprising observation in the pike bone was that the damage imprint is oval, extending orthogonal to the main collagen fiber orientation. This is clear indication that damage spreads in a way that is strongly affected by the texture. Preferential photoelectron ejection by the horizontal polarization of the synchrotron might also play a role, as previously suggested[14], though this was not seen in our experiments of other bony tissues. Due to the thin plate-shaped cAP crystals, photoelectrons are more likely to escape orthogonal to the fibril texture, and therefore in the homogeneous and less dense pike nanocomposite they propagate further than in higher-density bones with more complex textures. Consequently, pike bone samples show a lateral extent of damage that is more anisotropic, an effect that is to be expected in similar highly-anisotropic bony tissues such as turkey tendon. As shown by Monte Carlo simulations provided in Supplementary Figure 7, electron penetration depth is related to cAP density. Therefore, textures in bone including both density and collagen organization variations, play important roles in the spread of damage by photoelectron scatter in X-ray irradiated samples. Note that very thin samples (a few μm thick) might allow photoelectrons to escape causing less damage than what is seen in thicker samples. This however is not likely to overcome lateral damage spread outside of the beam path. Radiation damage to collagen is of little consequence to conventional μCT imaging of bone, since such damage has no influence on the resulting 3D reconstructed data, due to the negligible effects on sample density. Accumulated damage following high-flux μCT imaging is visible in our samples by SHG and is likely to only become a problem with very high-resolution nanoCT imaging, where sample motion due

to strain release will create artifacts that affect reconstructions. X-ray imaging of small bone samples is thus probably not limited by X-ray resolution, but rather by the accumulation of damage during measurement. In fact, after irradiation with high doses, structural integrity may be lost yielding spontaneous crack propagation, as shown in Supplementary Fig. 10.

This study showed that primary radiation damage swiftly develops in the collagen backbone of bones, and that damage is to be expected in all X-ray irradiated samples, which may call for reconsidering sterilization procedures applied to bone allografts or similar samples that require collagen integrity. Our observations demonstrated that damage is not strictly dose-dependent and therefore cannot be predicted simply based on classical calculations of dose as is used in living tissues. Instead, damage accumulates faster at the onset, and the rate of damage accumulation slows down with increased exposure, releasing dehydration-induced residual compressive strain in the mineral crystals. Though our data focuses only on dry bones, damage in wet bones is likely unavoidable in the same manner due to excitation of similar photoelectron scatter mechanisms, though absorption by water molecules is likely to protect the tissue. In hydrated samples, hydrolysis may change the distribution of damage, and is likely to contribute to cross-linking. We observe that the volume affected by damage exceeds the incident lateral beam size due to photoelectron scattering, amplifying the effects of the incident beam while creating larger domains of damage across the bony nanocomposite. In this context, cAP is an indirect but main contributor to the development and spread of radiation damage in bone.

Our study utilizes second harmonic generation (SHG) confocal laser scanning microscopy to directly visualize radiation damage in collagen in situ. This method is particularly suitable to identify damage to the collagen backbone because the SHG signal[49–51] arises directly from the collagen molecules that comprise the fibers in bones. Our results are of fundamental importance for predicting damage to bone that is analyzed or sterilized by ionizing radiation, and they identify Ca and other elements in the cAP nanocrystals as main sources of primary radiation damage due to unavoidable characteristic photoelectron excitation following X-ray absorption.

## Methods
### Sample preparation
Bone samples (Fig. 1a) were extracted from the cleithrum bones of 15 different recently captured pike fish, *Esox lucius*, purchased in a local fish market. In total, 64 pike samples were prepared. Of these, 25 pike bone samples were dried in an oven for 1 h at 120 °C one day prior to X-ray experiments, for different measurements as described below. Pigs jaws ($n = 3$) and bovine teeth ($n = 2$) were taken from a Berlin slaughterhouse providing meat for human consumption. Mouse bones ($n = 1$) were taken from carcasses of culled animals discarded as part of routine maintenance of the Charité animal facility mouse colony. Each of the bone samples was cut into elongated parallelepiped specimen using a slow speed precision water-cooled saw (IsoMet, Buehler, ITW Test and Measurement GmbH, Düsseldorf, Germany) employing diamond wafering blades. All samples were ground flat with silicon carbide grinding paper (1200 4000 grit) yielding ~500 × 300 × 3000 μm slices. 39 of the pike bone samples were stored in 70% ethanol and scanned with μCT in air (Fig. 1a). Three pike bones were demineralized: initially dehydrated in an alcohol concentration series of 25%, 50%, 75%, 80%, 90% and twice 100% for 30 min each. Following overnight storage in silica-gel, the samples were immersed in 17% EDTA solution for 24 h. These 3 demineralized samples, 2 pikes samples, 2 bovine teeth, 3 pig-jaws, and 1 mouse tibia were dried in an oven for 1 h at 45 °C one day prior to X-ray experiments. Following X-ray exposure, the samples were stored dry in sample membranes for several days, after which they were immersed in water immediately prior to SHG imaging.

## Measurement strategy

Each sample was measured by SHG before and after exposure to synchrotron radiation, selected samples were also measured by SEM-BEI to visualize the radiation damage as an imprint. During the irradiation experiments, each sample was mounted on stages in the different beamlines for either diffraction (XRD) or radiography experiments. The XRD measurements were performed either as 2D maps of non-overlapping points or as repeated line scans combined with rotation for XRD-$\mu$CT scans.

## X-ray irradiation experiments

**Synchrotron-based $\mu$CT.** Synchrotron-based $\mu$CT experiments were carried out in absorption and phase contrast[57] modes on beamline ID19 (ESRF, Grenoble, France). An indirect detector was used (OptiquePeter, France) with a pco.edge camera (PCO AG, Germany) to reach an effective pixel size of 0.65 $\mu$m. Per tomographic scan, 4000 projections were collected, using experimental settings as listed in Supplementary Table 1: $\mu$CT (17)–(19).

**Synchrotron-based X-ray diffraction (XRD) and XRD-microcomputed tomography (XRD-$\mu$CT).** Synchrotron-based XRD and XRD-$\mu$CT measurements were performed at the mySpot beamline of the BESSY II synchrotron light source (HZB – Helmholtz-Zentrum, Berlin, Germany)[58]. A Dectris M9 Eiger detector with an image size of 3269 × 3110 and a lateral pixel size of 75 $\mu$m was used. To acquire the elemental distributions, fluorescence was measured using a silicon drift detector (40 mm$^2$ SiriusSD, SGX Sensortech). A list of the various experimental settings used is given in Supplementary Table 1: XRD (1)–(15) and XRD-$\mu$CT (16). The sample-to-detector distance was kept constant (~345 mm), calibrated by measurements of alumina ($Al_2O_3$). For the experiments (6)–(9) and (15)–(16) the pinhole-sample distance was ~10 cm. For all other XRD-related experiments, the pinhole-sample distance was ~2 cm (see Supplementary Fig. 1). Each sample was measured in different non-overlapping points by moving the sample both laterally and along the long sample axis repeatedly rotating the sample with an exposure time of 15 s. XRD-$\mu$CT measurements were performed scanning lines of 33 points each, with a lateral step size of 25 $\mu$m and rotated from 0 to 360° with 42 rotation steps with a step-size of 8.57 degrees. At each point an XRD patter was recorded so that a total of 1419 diffraction patterns were collected. The beamline design is as follows:

X-rays for the mySpot beamline are provided by a 7 T wavelength shifter, which allows for hard X-rays experiments at the rather low-energy storage ring. The broad-spectrum (white) beam is focused by a toroidal mirror located 13 m from the source to a focal spot ~32 m from the source, providing almost 1:1 focusing and very low divergency. No additional focusing elements were used, in order to conserve the low divergence and make the beam profile determination more accurate. The FWHM profile of the beam at focus is plotted in Supplementary Fig. 1. The exact divergence at the sample position was calculated using dedicated ray tracing software[59] on a digital copy of the beamline and was estimated to be 0.23 mrad horizontally and 0.045 mrad vertically. Scans across sharp tungsten knife-edges confirmed these results experimentally.

After focusing, the beam is monochromatized using a Mo/B4C multilayer monochromator. The monochromator has the bandwidth of $E/\Delta E = 500$ providing an order of magnitude brighter beam, compared to a crystal monochromator, and the bandwidth is still narrow enough to provide sufficient resolution for diffraction analysis. The beam was shaped using circular platinum pinholes of various diameters (Supplementary Table 2). Intensity of the beam is measured using a calibrated photodiode mounted after the sample. This diode was also used to measure the transmission of the sample or the intensity of the primary beam. The primary beam is assumed to have a circular profile with constant intensity convoluted with a Gaussian function leading to some smearing with increasing distances from the pinhole (Supplementary Figure 1).

## Second harmonic confocal laser scanning microscopy

Second harmonic generation (SHG) imaging was performed within 10 days after sample irradiation using a confocal laser scanning microscopy (CLSM, Leica TCS SP5II confocal microscope, Leica Microsystems GmbH, Wetzlar, Germany). The SHG signal was generated using a Spectra Physics Ti:sapphire laser (Mai Tai HP, Spectra Physics, Santa Clara, CA, USA) with 100 fs pulse width at 80 MHz and wavelength of 910 nm. Water immersible objective lenses 40.0× and 25.0× with numerical apertures of 1.1 were used with a pinhole of 600 $\mu$m. The effective pixel size was ~600 nm with a voxel depth of ~1 $\mu$m. Laser power and detection parameters were kept constant for all experiments.

## Scanning electron microscopy

Backscatter electron imaging (SEM-BEI) was performed with a Phenom-XL scanning electron microscope (Phenom-World, Eindhoven, Netherlands) using an acceleration voltage of 20 kV in the low vacuum 60 Pa mode, employing a working distance of 16 mm.

## Quantification of radiation damage: experimental data

**Effects on collagen.** The highly organized collagen protein backbone results in delocalized electrons with strong electric dipole moments. This non-centrosymmetric environment provides the SHG signal origin, where regions of coherent summation of the responses from many triple helices align in the same orientation[60]. In SHG microscopy the signal is dependent on size and packing of the molecules[61,62]. Bancelin et al.[60] reported a sensitivity threshold of SHG microscopy and found a fibril diameter of 30 nm with $1.2 \times 10^6$ peptide bonds. The SHG signal scales quadratic with the density of collagen triple helices aligned in parallel in the focal plane. If this plane is oriented perpendicular to the incident laser beam, an SHG signal is generated while a parallel orientation leads to an annihilation of the SHG signal in back scattering imaging mode. SHG signals of the collagen in bones are detectable with a wavelength range of $\lambda = \sim450$–460 nm[50] with the main signal emerging up to 150 $\mu$m into the matrix of mineralized collagen fibers. In regions damaged by irradiation, the intensity of the SHG signal is diminished or absent[49–51].

SHG images of collagen before and after irradiation were quantified using Fiji[63] in the form of 3D stacks of images. The boundaries of affected regions of the bones were identified using the Z-project function to produce maximum intensity and standard deviation 2D images. The damaged domain was defined as the area with reduced intensity as measured at the full width of profiles along and across each XRD measurement point.

**Effects on mineral.** Six pike bone specimens, dried at 120 °C in an oven for 1 h, one mouse tibia, two bovine teeth, and 2 pig jaw bones dried at 45 °C in an oven for 1 h were scanned by XRD at multiple spots on lines across each sample. Each sample was irradiated at three non-overlapping spots with distinct fluxes, while a diffraction pattern was collected every 10 s. Details can be found in Supplementary Table 1: XRD (7)–(14). The XRDUA software package (version 6.4.3.2[64]) was utilized for experimental setup calibration and peak profile integration of (002) Debye ring of the mineral nanoparticles. A mask was applied on the diffraction patterns to disregard vertical and horizontal stripes of the EIGER 9M detector during the 360 degrees range azimuthal integration procedure. Diffraction profiles were fitted with a combination of a Voigt and linear function using Python 3.7 and the LMFIT package[65]. Profiles were corrected by subtraction of an empty beam prior to fitting for quantitative analysis in the case of the X-ray diffraction experiments.

## Quantification of radiation damage

**Monte-Carlo simulations.** Monte-Carlo simulations of electron trajectories in mineral were performed with the Casino modeling tool v2.51(2.5.1.0)[66]. A substrate of 300 μm was constructed with the constituents of mineral ($H_2O_{26}P_6Ca_{10}$ and a density of $1.01\,g\,cm^{-3}$), collagen ($H_{0.49514}C_{0.31554}N_{0.08738}O_{0.10194}$ and a density of ~$0.41\,g\,cm^{-3}$) and bone ($Ca_{0.09863}P_{0.05918}O_{0.31413}N_{0.04946}C_{0.17861}H_{0.3}$ and a density of ~$1.41\,g\,cm^{-3}$). Due to the difference between incoming (18 keV) and Ca-K shell XRF emission energies, electron energies 13.9619 keV were chosen and 2000 electron pathways were simulated within the fish bone sample with a beam radius of 5000 nm.

## Reporting summary

Further information on research design is available in the Nature Research Reporting Summary linked to this article.

## Data availability

The data supporting the findings of this study are available from the corresponding author upon request.

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

## Acknowledgements

The authors thank BESSY II (HZB- Helmholtz-Zentrum, mySpot beamline, Berlin, Germany) and the ESRF (European Synchrotron Radiation Facility (ID19), Grenoble, France) for providing beamtime. We thank Ansgar Peterson of the BIH-Julius Wolff Institute and SFB 1444 for providing access to SHG microscopy. We further acknowledge Wout de Nolf for providing the XRDUA framework for XRD analysis as well as Jens Poloni and Andre Saschenbrecker from the Berliner Fishmarket for sample acquisition. This research is partially supported by the German Israeli Foundation (GIF) grants I-1278, I-1496 and by student grants of the Berlin-Brandenburg School for Regenerative Therapies (BSRT), Berlin, Germany, and the Minerva foundation. J.-B.F. acknowledges support by the US Dept. of Energy (contract No. DE-AC52-07NA27344). We thank the SFB 1444 for partial open access funding.

## Author contributions

P.Z., I.Z., and K.S. designed the study, K.S. performed SHG and SEM-BEI measurements. K.S., P.Z., I.Z., and A.R. performed XRD, XRD-$\mu$CT, and $\mu$CT measurements, J.-B.F. performed XRD analysis, K.S. created all plots and figures, K.S. and E.S. performed Monte-Carlo simulations, I.Z. and E.S. performed beamline simulations, E.S., K.S., and P.Z. created the damage model. Authors Paul Zaslansky and Ernesto Scoppola equally contributed to this work.

## Funding

## Competing interests

The authors declare no competing interests.
