## [Peer Review File · Nature Communications]

Primary radiation damage in bone evolves via collagen destruction by photoelectrons and secondary emission self-absorptionREVIEWER COMMENTS

Reviewer #1 (Remarks to the Author):

In their paper "Ca-fluorescence and self-absorption drive primary radiation damage in bone collagen" the authors show that the main driver of degradation in the collagen network of mineralized tissue is a cascade of photo-electron excitations driven by x-ray fluorescence emitted by the mineral portion of the tissue.

They use a combination of x-ray scattering and imaging studies to induce radiation damage and then use either the strain extracted from the x-ray diffraction signal or the signal from the remaining collagen by second harmonic generation to estimate the effect of radiation damage. From the way the x-ray damaged zone is propagating through the illuminated volume, the authors estimate a typical propagation length of the damage for the chosen mineralized tissue, which is in the order of $\sim 10 \mu\text{m}$. As this propagation effect is decoupled from the incident beam size, they claim that it is of high relevance, in particular for x-ray micro- and nanobeam studies where the absorbed x-ray dose is not an accurate descriptor of sustained damage any longer and has implications for the current practice of sterilizing bone allografts by x-ray and gamma radiation.

The question the authors are attempting to address is of relevance to the field and the way the authors try to address it by combining x-ray diffraction and SHG imaging is indeed an original and promising approach.

That said, I find it difficult to follow the paper, appreciate the methodology they have used and follow their reasoning to come from the experimental data to the conclusions. I have a couple of major points that I will address here before I go into more minor details:

In general, I find that the choice of experimental parameters as well as the description of the experimental details leaves something to be desired. As the authors rightly point out, the deposited dose might not be the most important factor. I am wondering why the authors chose exposure times which are impractically long (8s or more) and deviate strongly from the few ms, (see e.g. Wittig et al 2019 DOI:10.1021/acsnano.9b05535 for a 50 ms exposure time diffraction tomography study), that are currently used for scanning diffraction experiments in the field. The reason why I point this out is because there is a whole body of literature on radiation damage in proteins where research has suggested to outrun the onset of radiation damage by staying with a few ms exposure times. See e.g. Owen 2012 DOI:10.1107/S0907444912012553

What I would like to see is an extension of parameter space towards lower, practically relevant exposure times ranging from the ms to the sub-s time scale at a high flux microprobe beam line like Petra III P06, ESRF ID13 or MaxIV Nanomax.

I would urge the authors to include a more in-depth review of radiation damage and its propagation. The field of protein crystallography has investigated radiation damage and its propagation in great detail and I miss some important work here. An overview to start with could be the work of Nave and Garman DOI: 10.1016/0969-806X(95)92800-E.

It is in fact well known that the propagation of radiation damage outside of the primary, illuminated volume is driven the cascading propagation of photo-electrons. It is regular practice to consider this effect in the modelling of radiation damage in proteins by the RADDOSE 3D package (Bury et al 2018, DOI: 10.1002/pro.3302).

I would like to point the authors to the work of Bras and Stanley 2016 (DOI: 10.1016/j.jnoncrysol.2016.06.020) or Bras et al 2021 (DOI:10.1088/1361-648X/ac1767) for an overview over radiation damage, specifically the spreading for several μm s outside of the irradiated volume by secondary photoelectron cascades. While I agree with the authors that the transfer from organic crystals to mineralized tissue is of relevance, the reported damage mechanisms that they observe here are in fact well known.

By the same token, I would suggest to consider the implementation of a more realistic model for their electron propagation model. It is current practice to use a two or three beam model as e.g. outlined by Holton et al 2007 or De la mora 2020 to have a more

realistic simulation. The way the authors appear to have currently modeled the beam interactions with the sample (e.g. visible in SI Fig 4 where the integration of dose leads to a top-hat like profile) is unrealistic as it does not take the beam profile into account and neglects the effect of the beam tails. Using the above suggested multi-beam methods will solve this and will lead to more realistic estimation.

With respect to the experimental description, I miss some very important aspects like the actual beam (not synchrotron source) divergence and a description of the size estimation procedure beyond the scant info provided by SI Fig3, a description of the flux estimation, monochromatization scheme, energy bandwidth, the optical elements involved in shaping and collimating the beam, the pinhole to sample distance. Figure 5 is suggesting that the incoming beam is actually convergent while the illuminated volume appears to be an oval channel with parallel sides.

I also completely miss the description of the XRD-CT experiment like step size, number of projections etc. In the current form, I would not be able to repeat the experiments.

In all the descriptions and figures, I take it that a parallel, oval beam profile is assumed, which I fear might be an oversimplification of the physical reality, especially as the authors stress the fact of time-dependent propagation of damage because this is exactly what weak beam tails would do.

The authors state on p.12/l 269 "Since the energies emitted by Ca and P in the crystals are fixed and independent of the incoming beam energy, collagen destruction and hence radiation damage is not directly related to bone mass."

I disagree with this statement and on the contrary as the amount of emitted as well as absorbed X-ray fluorescence is linearly dependent on the amount of mineral present in the sample, thus the bone mass.

I would thus ask the authors to present more data that shows that their observations hold true for other biomineralized tissue. I would suggest to look into enamel as a highly mineralized tissue and mouse or rat bone as an accepted 'medium mineralization' model system which is medically relevant. Alternatively, mineralized turkey leg tendon does present a gradient of mineralization which could be explored further. While I see the point of the authors of structural homogeneity of the chosen pike bone, I am wondering whether this is truly a contributing factor for a beam of ~25 um diameter size. Most of the structural features are well below the size of the beam anyway and should be averaged over the beam cross section and the substantial sample thickness of 300 μm. The chosen model system presents a rather low degree of mineralization and is rather exotic in my opinion which makes it more difficult to draw general conclusions from it.

While the authors nicely explain the underlying contrast mechanism of SHG, I would like to see this method used to its full potential to generate a proper quantification of the collagen degradation beyond determining the size of radiation-damaged area. From a properly calibrated setup, the decay of the collagen signal should be quantifiable.

Another parameter that I miss is the description of the sample treatment between the x-ray exposures and the SHG measurements. What was the time delay before the SHG measurements? Where measures taken to keep the sample from absorbing moisture, which would alter the propagation of radiation damage post x-ray exposure.

Some minor points:

For the fit presented in Fig 4, I am wondering about the physical reasons to used first order exponentials to fit the data.

Line 143: I am not sure if elaborate is the right choice here? I would think 'pronounced' makes more sense in the context

Line 439: There appears to be a point missing in the python version.

SI Fig 5: What is the meaning of the three different colours (red, yellow, blue) for the electron trajectories?

In summary, I am convinced that the authors have identified an interesting question of importance to the field, especially with regards to the current upgrades of synchrotron facilities and I like the fact that they combine x-ray diffraction and imaging with SHG imaging to determine the amount of structural damage to collagen.

I however feel that their chosen experimental parameters do not reflect the experimental reality of today, the lack of knowledge about the experimental details, like realistic beam parameters does not allow me to find full support for their claims in the experimental data. The same applies to their modelling attempt. The model appears too simplistic in terms of beam modelling to convince me and support the claims of the authors.

I am convinced that the mechanism the authors describe is in principle correct, as it is fully in line with the radiation damage literature on protein crystals and agrees well with the anecdotic observation on damage propagation made during experimental sessions, but it would need a thorough revision to align the claims made by the authors with the experimental data they actually present and this report might be better placed in journal with a more x-ray focused readership.

Reviewer #2 (Remarks to the Author):

This is an important paper. The authors demonstrate that damage in bone occurs not only by the direct action of the incident beam photons but also by a cascade of secondary events. This is something that the referee has not seen addressed in the literature. As the referee routinely considers dose effects, the referee doubts that such a study has appeared before the present paper. There are a couple of things the authors should do or should discuss before the paper is published.

The authors go through a lengthy and intrinsically imprecise calculation of the attenuation of the beam and the bone density. The value they come up with is rather low from what the referee expects. Maybe the bone in the pike really is low density. The authors make a big point of microtomography scans throughout the paper. Why didn't they use the mean linear attenuation coefficients (LAC) that come from synchrotron microtomography (which they performed)? Comparison of the experimental LAC with a calculated mass attenuation coefficient for the energy (or a tabulated reference value like that from the NIST Tables for cortical bone reference material) would give a much better value of density. One would hope that this would exactly match the transmissivity determination. The referee regards this as an essential addition to the paper, but one that is relatively minor to implement. There are sure to be chemical composition effects, but given that Ca is the predominant absorber (by a lot), these effects do not matter for LAC-based determinations.

The authors should have provided a reference value of the c-lattice parameter for the wet bone. That way the reader can quantitatively verify the authors contention that there are, in fact, drying related macrostrains.

Further, what are the initial (deviatoric) macrostrains in the wet samples (and in the dried samples)? Although the c-reflection rings look to be incomplete, one expects that there is enough intensity to get an estimate from the patterns. Could the authors please provide this information; it is quite straightforward.

What is the crystallite size of the bone (Scherer Eq.)? It is true that, from only one order of c-reflection, one cannot separate microstrain effects, nonetheless, the authors should give this value and note that microstrain would produce an actual value of size that is somewhat larger. This value, while tangential to the main focus of paper, is important to

building up our understanding of values of apatite lattice parameters across Animalia.

In the referee's opinion, the authors should have demineralized one of the bone sections and checked the results of the beam damage. There are some very mild acid solutions that are quite effective. If the mineral were absent, then the damage as seen by SHG would accumulate very differently. There would, of course, be some disruption of the collagen but this should be minor. This could be discussed as future work.

The authors missed the following relatively common synchrotron diffraction "trick" to getting good diffraction patterns while limiting beam-induced damage. Gallant et al. in Bone 61 (2014) 191, for example, recorded multiple patterns at different time points from adjacent positions, never exposing the bone to more than one exposure. In the present context, one could investigate at quite small doses by using a similar strategy. One records, say, nine patterns from different nearby locations and adds the diffraction patterns. As a zero order approximation with a reasonably good area detector, the signal to noise improvement would approach a factor of three and lattice parameter prediction at extremely low doses could be obtained.

The use of the word "spread", in the sense of damage spread from the center of the beam, is extremely misleading, and the authors should really change their phrasing. The damage is indeed highest at the beam center and tapers off toward the edges of the beam and outside the beam. This is strictly a geometrical effect with the greatest concentration of Ca and P fluorescence (and ejected electrons) being at the center of the beam. It is quite likely that an unwary reader would take spread to mean something like the damage nucleated at the center of the beam and then spread out from that point. This must be taken care of, in the referee's opinion.

The referee noticed a few typos line 401 "is" should be "If"? somewhere there was a transposition of letters in "bone".

Answer to the reviewers

August 20, 2022

Submission of revision to Nature Communications

Nature Communications manuscript *NCOMMS-21-46473* **”Primary radiation damage in bone: collagen destruction by photo-electrons and secondary emission self-absorption”**.

A list of our corrections, point-by-point answers to the reviewers and edits is provided below. Reviewer comments appear in BLACK and our answers appear in BLUE with which we have also marked edits in the text.

Reviewer #1 (Remarks to the author):

In their paper “Ca-fluorescence and self-absorption drive primary radiation damage in bone collagen” the authors show that the main driver of degradation in the collagen network of mineralized tissue is a cascade of photo-electron excitations driven by x-ray fluorescence emitted by the mineral portion of the tissue.

Thanks to the extensive reviewer comments, we repeated and expanded our experiments, leading us to revise both the title and almost all of the text. We believe that following our extensive revision, we now better address the broader topic, including our previously missing assessments of the damaging role of the photoelectrons.

They use a combination of x-ray scattering and imaging studies to induce radiation damage and then use either the strain extracted from the x-ray diffraction signal or the signal from the remaining collagen by second harmonic generation to estimate the effect of radiation damage. From the way the x-ray damaged zone is propagating through the illuminated volume, the authors estimate a typical propagation length of the damage for the chosen mineralized tissue, which is in the order of 10 μm . As this propagation effect is decoupled from the incident beam size, they claim that it is of high relevance, in particular for x-ray micro- and nanobeam studies where the absorbed x-ray dose is not an accurate descriptor of sustained damage any longer and has implications for the current practice of sterilizing bone allografts by x-ray and gamma radiation. The question the authors are attempting to address is of relevance to the field and the way the authors try to address it by combining x-ray diffraction and SHG imaging is indeed an original and promising approach.

We thank the reviewer for these kind words.

That said, I find it difficult to follow the paper, appreciate the methodology they have used and follow their reasoning to come from the experimental data to the conclusions.

We have extensively revised the paper, providing more detail and a large range of new experiments that significantly improved the results and therefore our predictions. We hope our reworked reasoning better explains how we reach our revised conclusions.

I have a couple of major points that I will address here before I go into more minor details: In general, I find that the choice of experimental parameters as well as the description of the experimental details leaves something to be desired.

We agree, and have therefore performed a large array of new experiments with different bones, different beam sizes, better control of the parameters. Further, we now provide quantifications for several of the observed effects which, coupled with additional electron microscopy, multiple simulations and deeper literature survey, have led us to refine our observations based in part on new modeling that we developed.

As the authors rightly point out, the deposited dose might not be the most important factor. I am wondering why the authors chose exposures times which are impractically long (8s or more) and deviate strongly from the few ms, (see e.g. Wittig et al 2019 DOI:10.1021/acsnano.9b05535 for a 50 ms exposure time diffraction tomography study), that are currently used for scanning diffraction experiments in the field.

The present work addresses the physical interactions between photons and bone material, which are somehow independent of the question of exposure time: the longer exposures simply produce effects that are easier to demonstrate and quantify. No doubt, shorter exposures may help solve problems related to measurements of bone by X-rays, but they are not likely for repeated tomographic scenarios and are anyway not the focus of this work. The scans we used are just examples (from real experiments) and serve to highlight the complexity and the difficulties that relate to photoelectron scattering from apatite crystals in bones. These effects become very significant the longer the bone is exposed, especially when tomography methods are used (requiring repeated exposure of the same volume). They are also usually undetected. In fact, often samples are repeatedly scanned with the likely result that damage will accumulate. We do not doubt that faster measurements are possible, though brighter sources and higher fluxes are then needed (not always available) which in turn may simply eject electrons at higher rates. We have checked shorter exposure times, but the weak effect on our measurement method of SHG are not quantifiable, perhaps future developments may resolve this. Our purpose is therefore to show that the ionization occurring has important effects, and longer exposure times help us make our case. Though we have not shown this explicitly for 50 ms exposure times, a subject for possible future research, we believe that our results are valid and of high importance.

The reason why I point this out is because there is a whole body of literature on radiation damage in proteins where research has suggested to outrun the onset of radiation damage by staying with a few ms exposure times. See e.g. Owen 2012 DOI:10.1107/S0907444912012553.

Interesting point: Cryo methods also help, but none claim nor attempt to prevent radiation damage, rather, they divert it. Overall, as now better cited in our revised text, the use of smaller beams for protein crystals goes hand in hand with using smaller protein crystals where photoelectron escape distributes and mitigates some of the radiation damage occurring directly in the X-ray beam path. But this can't work for reasonably sized bone samples (far larger than the beam used): in bone that is far denser than most organic crystals, our data show that the energy is redistributed into adjacent bone regions that are typically of interest for consecutive additional measurement. We clearly show now, that in adjacent regions up to several micrometers wide, the bone matrix becomes extensively damaged, far from the point irradiated with X-rays.

What I would like to see is an extension of parameter space towards lower, practically relevant exposure times ranging from the ms to the sub-s time scale at a high flux microprobe beam line like Petra III P06, ESRF ID13 or MaxIV Nanomax.

We have now performed measurements with a range of beam sizes and have developed a model that we use to predict damage effects deep into the nanometer range of beams based on extrapolation. We fully understand that progress is being made in a range of beamlines that make fast measurements to try and reveal structural integrity prior to radiation disintegration. Yet, no ionizing radiation method is able to avoid the damage, and therefore an important question is how to at least know about it and possibly account for it. This is particularly true for the case of bone, where high energy electrons are excited due to absorption and ionization, which cannot be avoided due to the physics involved. There may come a time where diffraction signals from bone can be captured faster than radiation damage accumulates, however, current common-practice realistic exposures to radiation either for sterilization, imaging or structural investigations typically include long exposure times, and in particular tomography measurements lead to significant damage in the protein fibers that are usually not considered when using such X-ray methods. The new revised measurements we performed span the 100 to 5 μm range, from which we extrapolate to smaller beams, based on the principals of photoelectron interactions that we now better cover. It is possible that there is value in adding confirmatory experiments with smaller beams and faster measurements, however especially after we performed extensive multiple measurements with a range of beam-sizes as described in this revised text, we believe that such an endeavor falls outside the scope of the present paper.

I would urge the authors to include a more in-depth review of radiation damage and its propagation. The field of protein crystallography has investigated radiation damage and its propagation in great detail and I miss some important work here. An overview to start with could be the work of Nave and Garman DOI: 10.1016/0969-806X(95)92800-E.

We have followed this advice and we also thank the reviewer for the reference that we previously overlooked, which has led us to additional important missing literature (Sanishvili et al, (2011), DOI: 10.1073/pnas.1017701108). We are however strongly limited by the scope of what we may review within this single work. We believe that our improved literature helps to better clarify the achievements in this work and that the experimental work can be better understood in the broader literature context.

It is in fact well known that the propagation of radiation damage outside of the primary, illuminated volume is driven the cascading propagation of photo-electrons. It is regular practice to consider this effect in the modelling of radiation damage in proteins by the RADDOSE 3D package (Bury et al 2018, DOI: 10.1002/pro.3302).

We agree, and we now better cover this topic. We had erroneously removed this reference from our original submission (keeping an earlier reference) due to restrictions in the number of references allowed, but now it is back. Note, however, that to date, none of the available simulation packages is able to reliably reproduce the complex interactions within the composite structure of bone, and therefore so far there are no reports of its use in predicting radiation damage in studies of collagen in native bone tissue. In that context, we believe that what is well known for proteins is actually not known to the bone community.

I would like to point the authors to the work of Bras and Stanley 2016 (DOI: 10.1016/j.jnoncrysol.2016.06.020) or Bras et al 2021 (DOI:10.1088/1361-648X/ac1767) for an overview over radiation damage, specifically the spreading for several μm s outside of the irradiated volume by secondary photoelectron cascades.

We are very grateful to the reviewer for this hint that we indeed overlooked in the original submission! As mentioned above, we now better review the topic which also helps us better explain our observations.

While I agree with the authors that the transfer from organic crystals to mineralized tissue is of relevance, the reported damage mechanisms that they observe here are in fact well known.

While we agree that the physics is all known, it has never been shown nor discussed in the context of irradiative handling/studies of bone. This is important to resolve, because bone is a material often irradiated and studied by X-rays and for which no previous work has ever fully resolved these mechanisms (we also only touch the surface of the problem). This is why we believe this topic is actually not properly appreciated by the bone research community, meriting our dedicated quantitative experimental validation and reporting. The reviewer is likely aware that the literature regarding radiation damage in bone by-and-large completely ignores the effects of photoelectrons and fluorescence, and hence the common practice of calculating/reporting radiation damage is limited to estimates of absorption, flux and mass. We hope the extensively revised text convinces the reviewer how important it is to link these 2 fields of research, as we have done in the revised manuscript.

By the same token, I would suggest to consider the implementation of a more realistic model for their electron propagation model.

We agree and have done so – we now have a new model, fully described in the revised text.

It is current practice to use a two or three beam model as e.g. outlined by Holton et al 2007 or De la mora 2020 to have a more realistic simulation. The way the authors appear to have currently modeled the beam interactions with the sample (e.g. visible in SI Fig 4 where the integration of dose leads to a top-hat like profile) is unrealistic as it does not take the beam profile into account and neglects the effect of the beam tails. Using the above suggested multi-beam methods will solve this and will lead to more realistic estimation.

We have removed SI Fig. 4 and have extensively revised our model and supplementary figures to provide a more realistic description of the electron and photons involved. After extensive checking and simulation, with lots of new measurements, we now have a better description of how our beam profile is to be taken into account, leading indeed to a more realistic estimation, as requested.

With respect to the experimental description, I miss some very important aspects like the actual beam (not synchrotron source) divergence and a description of the size estimation procedure beyond the scant info provided by SI Fig3, a description of the flux estimation, monochromatization scheme, energy bandwidth, the optical elements involved in shaping and collimating the beam, the pinhole to sample distance.

We agree and have now included a comprehensive description. Specifically, we now also include new SI figures and tables to highlight the pinhole- and beamline configurations. All these missing aspects have been added to the methods and to the extensively enhanced supplementary sections.

Figure 5 is suggesting that the incoming beam is actually convergent while the illuminated volume appears to be an oval channel with parallel sides.

We agree. Our schematic figure, currently figure 7, is completely revised to better represent the beam geometry. We also included a new supplementary figure of the beamline configurations.

I also completely miss the description of the XRD-CT experiment like step size, number of projections etc. In the current form, I would not be able to repeat the experiments.

We have now provided the previously missing experimental details.

In all the descriptions and figures, I take it that a parallel, oval beam profile is assumed, which I fear might be an oversimplification of the physical reality, especially as the authors stress the fact of time-dependent propagation of damage because this is exactly what weak beam tails would do.

We agree, and, further to later recommendations of this reviewer, we have completely repeated the experiments and measured a range of new samples (now reported in new figure 3), where we used a short pinhole-to-sample distance (configuration 1 in supplementary figure 1) essentially devoid of tails, in addition to other measurements with a beam profile that includes marked tails (configuration 2).

The authors state on p.12/l 269 "Since the energies emitted by Ca and P in the crystals are fixed and independent of the incoming beam energy, collagen destruction and hence radiation damage is not directly related to bone mass." I disagree with this statement and on the contrary as the amount of emitted as well as absorbed X-ray fluorescence is linearly dependent on the amount of mineral present in the sample, thus the bone mass.

We have rephrased the text to better reflect what we mean, we now also think the original assertion was not correct. The rate of damage accumulation is what we find to be non linear, as now better explained in the text.

I would thus ask the authors to present more data that shows that their observations hold true for other biomineralized tissue.

We have reworked and have now added a range of collagen-based mineralized tissues including some demineralized samples, for which SHG imaging can be used.

I would suggest to look into enamel as a highly mineralized tissue

It is indeed highly mineralized, but irrelevant for our focus, since it is not a bony nanocomposite of apatite and collagen. We therefore chose other samples as described.

and mouse or rat bone as an accepted 'medium mineralization' model system which is medically relevant. Alternatively, mineralized turkey leg tendon does present a gradient of mineralization which could be explored further.

We have now included: mouse bone, bovine teeth and pig jaw as a selection of relevant bony tissues with well described mineralized collagen fiber nanostructures. These new data confirm our observations and help us better understand the interactions that we report.

While I see the point of the authors of structural homogeneity of the chosen pike bone, I am wondering whether this is truly a contributing factor for a beam of 25 μm diameter size. Most of the structural features are well below the size of the beam anyway and should be averaged over the beam cross section and the substantial sample thickness of 300 μm . The chosen model system presents a rather low degree of mineralization and is rather exotic in my opinion which makes it more difficult to draw general conclusions from it.

These are important aspects that we believe increase the importance of this work. The pike bone is on the one side homogeneous (lacking osteocyte and similar inclusions) but on the other side also low density bone (similar to healing bone) and also highly anisotropic, similar to the important 'mineralized turkey tendon' model. The reviewer makes the case that the structural features are much smaller than the beam that we use. However, they are much larger than the scattered electrons which our data shows to cause damage extending many microns away from the origin. We agree that our initial submission made it difficult to draw general conclusions and hence the excellent idea to add other bones has helped us here. Critically we observe that (a) different bones with different densities suffer different magnitudes of damage. We believe that this needs to be discussed by the community. Though higher density bones generate more electrons, they are denser, and hence travel shorter distances (see our new Monte Carlo simulations). (b) fiber layouts also have a critical effect, since electrons will have much higher probabilities to scatter laterally, orthogonal to the main axis of mineralized fiber. This means that local texture will affect the spread of primary radiation damage. By lacking internal inhomogeneities and surfaces such as osteocytes that scatter electrons, we are here able to better understand the damaging effect that photoelectrons have. We agree that our pike might appear to be exotic in that respect, but we now believe that by comparison to other bones, it helps draw general conclusions better than in the earlier submission. In fact, it may also help better understand anisotropic bony systems such as the mineralized turkey tendon, where, similar to pike, the collagen fibers are highly co-aligned.

While the authors nicely explain the underlying contrast mechanism of SHG, I would like to see this method used to its full potential to generate a proper quantification of the collagen degradation beyond determining the size of radiation-damaged area. From a properly calibrated setup, the decay of the collagen signal should be quantifiable.

We have attempted this, but due to illumination and imaging constraints inherent to two-photon confocal laser scanning microscopy of collagen fibers, this is not currently possible, and must wait for future work. The method is needed and used to trace the expanding volume of radiation damage in a variety of bony tissues.

Another parameter that I miss is the description of the sample treatment between the x-ray exposures and the SHG measurements. What was the time delay before the SHG measurements? Where measures taken to keep the sample from absorbing moisture, which would alter the propagation of radiation damage post x-ray exposure.

We have now added this information, but remind that the bones were measured dry, after gentle heating in an oven to remove all the free water. Thus slower processes of radical damage were minimized. In fact, repeated imaging of the same bones by SHG revealed that there was no change in the lateral dimensions of damage due to storage conditions.

Some minor points:

For the fit presented in Fig 4, I am wondering about the physical reasons to used first order exponentials to fit the data.

The figure has radically changed based on new data. In the extensively revised supporting information section, we now also provide an explanation for why first order exponentials are good choices to fit the data (especially the new Fig. 4).

Line 143: I am not sure if elaborate is the right choice here? I would think 'pronounced' makes more sense in the context

Changed.

Line 439: There appears to be a point missing in the python version.

Indeed, this incorrect reference was removed from the revised paper.

SI Fig 5: What is the meaning of the three different colours (red, yellow, blue) for the electron trajectories?

We are sorry that we neglected to explain this originally: we have now added legends to show that these are 'backscattered', 'primary' and 'secondary' electrons in the simulations.

In summary, I am convinced that the authors have identified an interesting question of importance to the field, especially with regards to the current upgrades of synchrotron facilities and I like the fact that they combine x-ray diffraction and imaging with SHG imaging to determine the amount of structural damage to collagen. I however feel that their chosen experimental parameters do not reflect the experimental reality of today, the lack of knowledge about the experimental details, like realistic beam parameters does not allow me to find full support for their claims in the experimental data. The same applies to their modelling attempt. The model appears too simplistic in terms of beam modelling to convince me and support the claims of the authors. I am convinced that the mechanism the authors describe is in principle correct, as it is fully in line with the radiation damage literature on protein crystals and agrees well with the anecdotic observation on damage propagation made during experimental sessions, but it would need a thorough revision to align the claims made by the authors with the experimental data they actually present and this report might be better placed in journal with a more x-ray focused readership.

We have extensively reworked the paper, completely revised both experiments and models, and we thus believe that our revised manuscript addresses the important constructive concerns that the reviewer has helped us identify.

Reviewer #2 (Remarks to the Author): _____

This is an important paper. The authors demonstrate that damage in bone occurs not only by the direct action of the incident beam photons but also by a cascade of secondary events. This is something that the referee has not seen addressed in the literature. As the referee routinely considers dose effects, the referee doubts that such a study has appeared before the present paper.

We thank the referee for the kind words: we also have never seen this topic addressed in the context of bone research. And now, following revision and building on all reviewer comments, and after we repeated and expanded our experiments, both the title and most sections of the text were extensively revised, better addressing the damaging role of high-energy photoelectrons.

There are a couple of things the authors should do or should discuss before the paper is published.

The authors go through a lengthy and intrinsically imprecise calculation of the attenuation of the beam and the bone density. The value they come up with is rather low from what the referee expects. Maybe the bone in the pike really is low density. The authors make a big point of microtomography scans throughout the paper. Why didn't they use the mean linear attenuation coefficients (LAC) that come from synchrotron microtomography (which they performed)?

Fish bones indeed have substantially lower mass and mineral densities (we now included references to this). Though we did perform microtomography (on ID19 of the ESRF, Grenoble, France), the high coherence and phase contrast enhancement make those LAC results rather unreliable for mineral density determination. Yet direct measures of attenuation were available for every XRD measurement point. The attenuation of the micro beams used for XRD were measured directly by determining photon counts both through and off the samples. From this we calculate the sample transmission, with no assumptions made about composition or rotation geometry. We therefore consider these to be at least as precise as any LAC measurement we could perform. Note also that, in different regions of the sample, density varies, so that there is no single 'correct' value. Still, the paper has now been expanded to include a range of other bony tissues, new Monte Carlo simulations of electron propagations in different bone composites as well as numerous XRD and other experiments that serve to strengthen and convince of the correctness of our results. We use tabulated values from the NIST tables to match the approximate composition of our samples.

Comparison of the experimental LAC with a calculated mass attenuation coefficient for the energy (or a tabulated reference value like that from the NIST Tables for cortical bone reference material) would give a much better value of density. One would hope that this would exactly match the transmissivity determination. The referee regards this as an essential addition to the paper, but one that is relatively minor to implement.

As mentioned above, the new results we now provide are indeed based on this reference data source, as now hopefully better clarified in the SI section (Supplementary Notes 3.1), note however, that cortical bovine and human bone are significantly denser than pike bone and hence we describe this evaluation in greater detail.

There are sure to be chemical composition effects, but given that Ca is the predominant absorber (by a lot), these effects do not matter for LAC-based determinations.

We agree, in fact we have tested small changes in Zn and Sr composition and found that the effects are indeed negligible. We now provide multiple simulations and discuss the role of the predominant Ca extensively.

The authors should have provided a reference value of the c-lattice parameter for the wet bone. That way the reader can quantitatively verify the authors contention that there are, in fact, drying related macrostrains.

Though we have a couple of test samples that we measured wet and dry, most of this study, comparing different bony tissues, focused on measurements in dry bones only, since we wanted to avoid secondary radiation damage effects due to radiolysis. In our revised work, we report newly collected data from several different bony tissues, including reference c-lattice values. For the diverse samples we now show strain relaxation due to irradiation which is actually somehow independent of the reference c-lattice parameter, as the reviewer may appreciate. In all cases, we get substantial strain relaxation with increasing exposure to X-ray. We believe that the data for wet samples falls outside the scope of this paper, and our wet test samples we found a c-lattice parameter of approximately 6.88 Å whereas the same samples after dehydration yielded a c-lattice parameter of almost 6.87 Å. We believe that in the interest of keeping the complex topic focused on primary radiation damage to bone, the emphasis on reporting dry bones only best serves to identify the main physical phenomenon that we report. Wet bones indeed deserve attention and have often been reported, therefore they fall outside the scope of this paper.

Further, what are the initial (deviatoric) macrostrains in the wet samples (and in the dried samples)? Although the c-reflection rings look to be incomplete, one expects that there is enough intensity to get an estimate from the patterns. Could the authors please provide this information; it is quite straightforward.

Our fish samples are highly anisotropic and indeed, the (002) reflection produces arcs, not rings, therefore there is not enough intensity to obtain deviatoric strain estimates. We have now included example diffraction patterns of the fish in the figures, to help convince of this. However, since the purpose of this work was to show the effects of accumulated damage for which the strains are mere 'sensors', we believe that the most significant effects of strain relaxation are observed by comparing the c lattice parameter of each XRD measurement to the initial state, which we show plotted over time in the revised figure 6. Nevertheless, we have provided in Supplementary Table 7, the results of c-lattice and other straightforward X-ray derived crystal characteristics, as requested.

What is the crystallite size of the bone (Scherer Eq.)? It is true that, from only one order of c-reflection, one cannot separate microstrain effects, nonetheless, the authors should give this value and note that microstrain would produce an actual value of size that is somewhat larger. This value, while tangential to the main focus of paper, is important to building up our understanding of values of apatite lattice parameters across Animalia.

At the request of the reviewer, we have now added estimates of microstrain fluctuations and crystallite sizes, based on 2 approaches: the Scherer equation and the Voigt deconvolution approach (following Forien et al, 2016). We indeed think that these values are a bit tangential to the main focus of the paper, which was not designed to importantly contribute to understanding apatite lattice parameter variations across different bones, rather to focus on possible damage due to radiation useage. One even wonders what sterilization of samples has doen to the literature reported values.

In the referee's opinion, the authors should have demineralized one of the bone sections and checked the results of the beam damage.

We completely agree! We now added this fascinating experiment which we did not think of initially and which we now couple to electron microscopy (backscatter imaging) to show the effects. The results of this experiment excellently contribute to our paper observations and insights. We have also added to the SI section, 2 XRF sum spectra to demonstrate the difference between the mineralized and demineralized samples. We should have thought about this earlier! Thank you!

There are some very mild acid solutions that are quite effective. If the mineral were absent, then the damage as seen by SHG would accumulate very differently. There would, of course, be some disruption of the collagen but this should be minor. This could be discussed as future work.

Indeed: we did not need any acid, only EDTA to remove the mineral, and the SHG (and comparisons with newly obtained SEM-BEI) indeed shows that the damage accumulates differently and to a smaller extent as compared with mineralized samples!

The authors missed the following relatively common synchrotron diffraction "trick" to getting good diffraction patterns while limiting beam-induced damage. Gallant et al. in Bone 61 (2014) 191, for example, recorded multiple patterns at different time points from adjacent positions, never exposing the bone to more than one exposure. In the present context, one could investigate at quite small doses by using a similar strategy. One records, say, nine patterns from different nearby locations and adds the diffraction patterns. As a zero order approximation with a reasonably good area detector, the signal to noise improvement would approach a factor of three and lattice parameter prediction at extremely low doses could be obtained.

We assume the referee is correct about this, however, the purpose of the current work was not to limit beam-induced damage, but to quantify and understand it. We set out to explore the role of secondary processes due to the common practice of using X-rays in a variety of bone-manipulation conditions, spanning sterilization to imaging. Rapid, cryo and other approaches will all minimize radiation damage. However the physical phenomenon of photoelectron scattering and related interactions reveals the increasing importance of understanding single X-ray exposures with respect to smaller beams used. An important aspect to consider is that even adjacent points that do not fall in the irradiation path may accumulate damage due to the ejected photoelectrons, suggesting that adjacent points that not sufficiently spaces several micrometers apart, are likely to be affected. Furthermore, for tomography measurements, measurements on different points are not actually possible, due to the requirements of repeated measurements from different orientations. We believe that we have better covered this topic in our revision.

The use of the word "spread", in the sense of damage spread from the center of the beam, is extremely misleading, and the authors should really change their phrasing.

We have revised this whole concept extensively. Though we do use the word in different contexts in the revised work, we note that damage does spread or grow outside of the beam path, and does this as a function of exposure to increasing electron bombardment.

The damage is indeed highest at the beam center and tapers off toward the edges of the beam and outside the beam.

Our new data show with more detail, how the beam profile imprint affects the bones of different bony tissues, expanding beyond the beam dimensions with increasing exposure time. Our new data from 2 different configurations and using 2 different beam profiles (new Supplementary Figure 1) helps better identify damage in the irradiated field, versus damage spreading outside the area

directly irradiated by our X-ray beams.

This is strictly a geometrical effect with the greatest concentration of Ca and P fluorescence (and ejected electrons) being at the center of the beam. It is quite likely that an unwary reader would take spread to mean something like the damage nucleated at the center of the beam and then spread out from that point. This must be taken care of, in the referee's opinion.

We agree and hope that our revised explanations make this clear. Specifically we have new simulations and a model and we show how the distribution of high energy electrons distribute differently in the matrix outside the beam path.

The referee noticed a few typos line 401 "is" should be "If"? somewhere there was a transposition of letters in "bone".

Noted and corrected.

REVIEWERS' COMMENTS

Reviewer #1 (Remarks to the Author):

I thank the authors for preparing a revised version of their manuscript. I appreciate the work the authors have put into reworking their manuscript.

The now retitled paper "Primary radiation damage in bone: collagen destruction by photoelectrons and secondary emission self-absorption" by Sauer et al. reports on collagen backbone degradation of x-ray irradiated bones, studied by second harmonic generation microscopy and x-ray diffraction. The authors observed a loss in SHG intensity and its damage profile as well as a change in the diffraction, subsumed as residual strain relaxation by the authors. From these observations and modelling work, they propose a model of radiation damage propagation in bone, based on the effects of induced x-ray fluorescence and photo-electrons ejected during the absorption process. I want to thank the authors for providing a more balanced overview over the radiation damage literature and including a larger array of bone tissue types. This certainly improves the experimental side of the manuscript. I also appreciate the work the authors put into the accurate description of their experiment.

In general, I am satisfied with the revised manuscript and feel that it just takes a few, minor revisions before it can be considered for publication.

I have a question regarding the newly introduced model and the conclusions which are drawn from this model.

If I understand the model correctly, the relative damaged volume presented in Fig 4 assumes a constant sample thickness. I find this assumption questionable for the case of 2D scanning experiments, which are one of the main applications discussed in the paper. Here one usually tries to match the sample thickness to the beam size to reduce projection effects. This usually leads to a sample thickness equal to the beam size, usually down to a the sub-micron level where sample preparation constraints become an issue. By assuming a constant sample thickness for the various beam diameters, the relative damaged volume is increased significantly. While the case of XRD-CT indeed has a larger and constant sample thickness, I would think that the vast majority of publications in the field of XRD studies on bone mineralization use 2D scanning approaches. I would thus ask the authors to consider coupling their sample thickness to the beam size and present more realistic damage estimates in the main text and in the accompanying Figure. I would however find it interesting if the current Fig 4 is retained in the SI for the XRD-CT case.

I also want to add that reducing the sample thickness provides a mean to help the photoelectrons escape from the sample. This should be discussed and used to give some context to the rather sensational damaged volume numbers provided by the authors, in particular as they are solely based on the extrapolation of their model.

I also don't understand the sentence on the expected damage volume for smaller beams (p.9 | 217). The authors talk about a 1000% damage volume increase for a 1 um beam and a damage area exceeding 7500% for a 0.1 um beam. Are the authors sure that the areal unit is correct?

In summary, I want to commend the authors on the additional work they are presenting in this resubmission. I am convinced that this paper, after addressing the few, pending issues, is able to provide significant, new and important insights into the radiation damage mechanisms occurring in bone.

Reviewer #2 (Remarks to the Author):

Referee 2 found this to be a responsive revision and recommends the paper be accepted as revised.

On another note:

Referee 1 commented that exposure times of 50 ms is what the authors should have been

modeling citing Wittig et al. 2019. Referee 1 has this totally wrong. The cited exposure is for a single position for a single projection. Wittig et al. used >250 projections for each reconstructed slice, so the actual exposure received by each voxel in the diffraction experiments was about 12.5 s. So the authors exposures are quite reasonable. Therefore, the comments of the Referee 1 after this point re: smaller exposure times are largely irrelevant. This referee does these kinds of experiments routinely and the modeled exposures are in line with what is needed.

Answer to the reviewers

October 8, 2022

Submission of revision to Nature Communications

Nature Communications manuscript *NCOMMS-21-46473* **"Primary radiation damage in bone evolves via collagen destruction by photoelectrons and secondary emission self-absorption"**.

Point-by-point answers to the reviewers and edits are provided below. Reviewer comments appear in BLACK and our answers appear in BLUE with which we have also marked edits in the text.

Reviewer #1 (Remarks to the author):

I thank the authors for preparing a revised version of their manuscript. I appreciate the work the authors have put into reworking their manuscript. The now retitled paper “Primary radiation damage in bone: collagen destruction by photoelectrons and secondary emission self-absorption” by Sauer et al. reports on collagen backbone degradation of x-ray irradiated bones, studied by second harmonic generation microscopy and x-ray diffraction. The authors observed a loss in SHG intensity and its damage profile as well as a change in the diffraction, subsumed as residual strain relaxation by the authors. From these observations and modelling work, they propose a model of radiation damage propagation in bone, based on the effects of induced x-ray fluorescence and photo-electrons ejected during the absorption process. I want to thank the authors for providing a more balanced overview over the radiation damage literature and including a larger array of bone tissue types. This certainly improves the experimental side of the manuscript. I also appreciate the work the authors put into the accurate description of their experiment. In general, I am satisfied with the revised manuscript

We appreciate this kind reviewer feedback, and we also believe that our extensive revision significantly improved the work.

and feel that it just takes a few, minor revisions before it can be considered for publication. I have a question regarding the newly introduced model and the conclusions which are drawn from this model. If I understand the model correctly, the relative damaged volume presented in Fig 4 assumes a constant sample thickness. I find this assumption questionable for the case of 2D scanning experiments, which are one of the main applications discussed in the paper.

Actually, we make no assumption about the sample thickness: we simply examine the ratio between the volume of material in which damage evolves and the volume of material directly irradiated by the beam. This is the relative damage volume. These two volumes will have the same thickness, which then cancels out in the ratio, for any given point in the sample, as long as absorption is small. Clearly for very thick samples, where the photons are fully absorbed, this assumption will not do. However, such measurements will also not produce any diffraction or imaging results and are therefore not considered in our work. As now hopefully better explained in the new revision to our text, the ratio of these volumes is identical to the ratio of the area of observed damage to the cross-sectional area of the impinging beam.

Here one usually tries to match the sample thickness to the beam size to reduce projection effects. This usually leads to a sample thickness equal to the beam size, usually down to a the sub-micron level where sample preparation constraints become an issue. By assuming a constant sample thickness for the various beam diameters, the relative damaged volume is increased significantly.

Actually not, as explained in the previous answer. The relative damage volume is largely independent of sample thickness. In our work we kept the sample thickness constant to make it easier for ourselves to report damage results for a range of beam diameters to explore and try to explain the evolution of primary radiation damage. We agree that one usually matches sample and beam sizes. We mainly note that the absolute distance that electrons will scatter in the bone is on the order of 4 microns in each direction, see supplementary Figs 7,8. This in fact does not nullify the benefit of use of smaller beams, it only suggests that it may make sense to space measurement points far enough, when one wants to avoid damage to the organic matrix.

While the case of XRD-CT indeed has a larger and constant sample thickness, I would think that the vast majority of publications in the field of XRD studies on bone mineralization use 2D scanning approaches.

We agree.

I would thus ask the authors to consider coupling their sample thickness to the beam size and present more realistic damage estimates in the main text and in the accompanying Figure. I would however find it interesting if the current Fig 4 is retained in the SI for the XRD-CT case.

We believe that our work shows very realistic damage estimates, and that it in fact lays foundation for a new series of continuation studies that can build on our observations to follow bone nanostructure anisotropy by exploring the distribution of damage. To this end, both the experiments and simulations show that damage spread is not dependent on sample thickness, and hence we believe that figure 4 should be retained, while we have improved descriptions of our parameters and the results in the legend.

I also want to add that reducing the sample thickness provides a mean to help the photoelectrons escape from the sample.

We tend to agree, though we have no indication that electron scatter along the beam path (essentially the sample thickness) has a significant contribution to the lateral spread of damage outside of the irradiated volume. Still, we cannot rule this out, and in fact, it is indeed very likely that extremely thin samples will suffer reduced damage from electrons. However, on the same chord, extremely thin samples will suffer far more sample preparation damage. Further, such samples are not always very interesting for X-ray tomographic measurements as they are very thin. They also come with other challenges such as a weak interaction volume and a danger of possible loss of context due to the small size and possible mounting difficulties.

This should be discussed.

We agree and have added this interesting point to our discussion.

and used to give some context to the rather sensational damaged volume numbers provided by the authors, in particular as they are solely based on the extrapolation of their model. I also don't understand the sentence on the expected damage volume for smaller beams (p.9 | 217). The authors talk about a 1000 % damage volume increase for a 1 um beam and a damage area exceeding 7500 % for a 0.1 um beam. Are the authors sure that the areal unit is correct?

We have noticed that indeed we were not sufficiently precise about when we discuss the beam diameter versus when we describe the area affected by damage. In this context, we have modified p. 9, lines 211-220 (pdf), (217-217: latex) to clearly refer to the beam diameter and the predictions of expansion of damage. We agree that the numbers that we found (now better depicted as factors, rather than %) can be seen as being sensational. However, they are purely logical: if we have a beam of 0.4 micron diameter, and because the electrons will certainly scatter in bone up to 4 microns away (even in air, as we have shown) the ratio of damage diameter to beam diameter will be $(4+4)/0.4=20$. If such a beam is used to measure large bone samples that have sub-mm to mm dimension, damage to protein will spread x20 times the size of the beam. Clearly, if larger beam sizes are used, the relative damage will be smaller, though electrons will still spread 4 μm outside of the beam path. This we think is non-intuitive and is the basis of this publication. We hope that our revised text now better explains all this.

In summary, I want to commend the authors on the additional work they are presenting in this resubmission. I am convinced that this paper, after addressing the few, pending issues, is able to provide significant, new and important insights into the radiation damage mechanisms occurring in bone.

We thank the reviewer for challenging these important aspects of our work which we believe we have now addressed fully.

Reviewer #2 (Remarks to the Author): _____

Referee 2 found this to be a responsive revision and recommends the paper be accepted as revised.

We thank the reviewer for helping us improve the work and reach the desired quality.

On another note:

Referee 1 commented that exposure times of 50 ms is what the authors should have been modeling citing Wittig et al. 2019. Referee 1 has this totally wrong. The cited exposure is for a single position for a single projection. Wittig et al. used >250 projections for each reconstructed slice, so the actual exposure received by each voxel in the diffraction experiments was about 12.5 s. So the authors exposures are quite reasonable. Therefore, the comments of the Referee 1 after this point re: smaller exposure times are largely irrelevant. This referee does these kinds of experiments routinely and the modeled exposures are in line with what is needed.